# Assessing intra- and inter-molecular charge transfer excitations in non-fullerene acceptors using electroabsorption spectroscopy

Sudhi Mahadevan [1,2,3], Taili Liu[4], Saied Md Pratik [5], Yuhao Li[6], Hang Yuen Ho[1,2,3], Shanchao Ouyang[1,2,3], Xinhui Lu [6], Hin-Lap Yip [1,2,3,7], Philip C. Y. Chow [8], Jean-Luc Brédas [5], Veaceslav Coropceanu [5], Shu Kong So[9] & Sai-Wing Tsang [1,2,3] ✉

Organic photovoltaic cells using Y6 non-fullerene acceptors have recently achieved high efficiency, and it was suggested to be attributed to the charge-transfer (CT) nature of the excitations in Y6 aggregates. Here, by combining electroabsorption spectroscopy measurements and electronic-structure calculations, we find that the charge-transfer character already exists in isolated Y6 molecules but is strongly increased when there is molecular aggregation. Surprisingly, it is found that the large enhanced charge transfer in clustered Y6 molecules is not due to an increase in excited-state dipole moment, $\Delta\mu$, as observed in other organic systems, but due to a reduced polarizability change, $\Delta p$. It is proposed that such a strong charge-transfer character is promoted by the stabilization of the charge-transfer energy upon aggregation, as deduced from density functional theory and four-state model calculations. This work provides insight into the correlation between molecular electronic properties and charge-transfer characteristics in organic electronic materials.

Organic heterojunctions composed of electron donor and acceptor materials have been a viable approach for dissociating the strongly bonded Frenkel excitons in organic photovoltaic cells (OPVs). Although a non-fullerene acceptor (NFA) was used in the first efficient OPVs[1], fullerene-based derivatives had been the key players since the mid-1990s due to their efficient exciton dissociation when combined with various donor materials. However, it was realized that the large voltage loss and limited optical absorption in

fullerene-based OPVs have hindered further improvements in the device power-conversion efficiency (PCE)[2-5]. Recently, several novel NFAs have demonstrated promising improvements in terms of both device performance and stability[6-8]. In 2015, Lin et al. reported an OPV using an A-D-A type NFA, ITIC, which delivered an encouraging PCE of up to 6.8%[9]. This triggered the community to re-visit the development of NFAs, as it offers the perspective of unlocking the limitations of fullerene-based acceptors to achieve higher optical

[1]Department of Materials Science and Engineering, City University of Hong Kong, Hong Kong SAR, PR China. [2]Centre of Super-Diamond and Advanced Films, City University of Hong Kong, Hong Kong SAR, PR China. [3]Hong Kong Institute of Clean Energy, City University of Hong Kong, Hong Kong SAR, PR China. [4]College of Physics and Electronic Information, Yunnan Normal University, Kunming 650500 Yunnan, PR China. [5]Department of Chemistry and Biochemistry, The University of Arizona, Tucson, Arizona 85721-0041, USA. [6]Department of Physics, The Chinese University of Hong Kong, Hong Kong SAR, PR China. [7]School of Energy and Environment, City University of Hong Kong, Hong Kong SAR, PR China. [8]Department of Mechanical Engineering, The University of Hong Kong, Pok Fu Lam, Hong Kong SAR, PR China. [9]Department of Physics and Institute of Advanced Materials, Hong Kong Baptist University, Kowloon Tong, Hong Kong SAR, PR China. ✉e-mail: saitsang@cityu.edu.hk

absorption and better energy-level alignment with the donor materials.

Recently, an A-DA'D-A type NFA, Y6, introduced by Yuan et al. in 2019, demonstrated a record high PCE of over 18% with reduced voltage loss in a PM6:Y6 bulk heterojunction (BHJ) OPV[10–13]. It is found that Y6 molecules are able to form extensive crystalline molecular packing due to their molecular conformation, high molecular rigidity, and the absence of out-of-plane side chains[14–19]. Understanding the mechanism for efficient charge generation in Y6-based OPV has thus attracted much research attention. It has been reported that BHJs with Y6 or its derivatives as acceptors can facilitate ultrafast charge transfer (CT) and exciton dissociation despite the negligible energy offsets at the donor/acceptor interface[20–23]. Mechanisms involving the electronic and exciton delocalization promoted by the distinctive π–π packing[14] of Y6 or its large molecular quadrupole moment[24] have been proposed to rationalize the higher photovoltaic performance of Y6. However, these two features are also observed in other less performing acceptors, including fullerenes[25]. Very recently, transient absorption spectroscopy (TAS) data have revealed the existence of Y6 inter-molecular excitations in both neat Y6 and blend Y6:PM6 films[21]. Moreover, it was suggested that these inter-molecular excitations due to their charge-transfer (CT) character facilitate the formation of short-lived free carriers even in neat Y6 films[26]. In most general cases, the low-energy excitations in molecular aggregates could contain contributions from locally (Frenkel) excited (LE) states and CT excitations. Thus, in Y6 dimers, there are two LE states ($\Phi_{LE1} = |M_1^* M_2\rangle$ and $\Phi_{LE2} = |M_1 M_2^*\rangle$) and two CT states ($\Phi_{CT1} = |M_1^+ M_2^-\rangle$ and $\Phi_{CT2} = |M_1^- M_2^+\rangle$). Electronic-structure calculations indicate that in Y6 crystals the CT excitations are located below their LE counterparts and that CT and LE states are strongly coupled (the related electronic couplings are about 100 meV)[26]. Therefore, according to density functional theory (DFT) calculations, the low-energy excited states in Y6 crystals can be represented as a linear combination (hybridization) of CT and LE contributions[26–29]:

$$\Phi_{LE-CT} = c_{LE}\Phi_{LE} + c_{CT}\Phi_{CT} \qquad (1)$$

where the $c_{LE}$ and $c_{CT}$ coefficients define the weights of CT and LE contributions to the dimer excited states.

To the best of our knowledge, despite the great interest in the nature of the inter-molecular excitations in Y6, a direct experimental verification of their CT character is still missing. Electroabsorption (EA) spectroscopy, based on the Stark effect, measures the change in optical transition energy under the perturbation of an electrical field. This change depends on the variations in dipole moment and polarizability of the transition states and is susceptible to shed light on the correlation between the CT character and fundamental material attributes. The EA technique has been widely used to study the excitonic properties of semiconducting materials[30–34]. According to the Stark effect, as depicted in Eq. (2), the energy of a state would change by $\Delta E$ under an electrical field $F$, as a function of its dipole moment ($\mu$) and polarizability (p)[35]:

$$\Delta E = -\mu F - \frac{1}{2}pF^2 \qquad (2)$$

where $\Delta E = E(F) - E(0)$, with $E(F)$ and $E(0)$ the energies of the state with and without application of an external electrical field, respectively. As illustrated in Supplementary Fig.1, the differences in dipole moment $\Delta\mu$ and polarizability $\Delta p$ between the ground and excited states can be obtained by measuring the change in optical absorption of an organic film under an electrical field. In the prototypical hydrogenic model, $\Delta\mu$ (related to the change in exciton radius) and $\Delta p$ (related to the change in electron delocalization) are correlated, such that an increase in $\Delta p$ will come along with an increase in $\Delta\mu$[36,37]. However, in the case of a many-body system such as an organic molecule, this correlation might not be valid as it depends on the details of the electron and hole wavefunction distributions, see below. Since organic materials usually have a small dielectric constant, strongly bonded Frenkel excitons are typically formed upon photoexcitation, which leads to only a little increase in the electron-hole separation in the excited state with respect to the ground state. Consequently, the EA spectral characteristics of organics are typically contributed by $\Delta p$[38–41]. On the other hand, in the case of an excited state with strong intermolecular CT character, the change a in optical absorption will be dominated by the contribution from $\Delta\mu$ due to the large increase in electron-hole separation. To date, only a few small molecular systems have been reported with their EA characteristics mainly contributed by $\Delta\mu$[42–45].

In this work, we performed an in-depth and systematic analysis of the CT properties of two archetypical NFA molecules, namely Y6 and ITIC. We are aiming to investigate the CT properties of intra- (single molecules) and inter-molecular (molecular aggregates) excited states in NFAs and their correlation with molecular properties such as change in the state dipole moment and polarizability upon photoexcitation. First, we fabricated solid-solution thin films to tune the molecular packing by dispersing different loading ratios of the NFA molecules in an insulating polymer matrix. The effect of molecular packing will be discussed first, in terms of the thin film optical absorption spectral characteristics and the corresponding GIWAXS results. The optical absorption spectra were analyzed using the Franck–Condon progressions to determine the contributions from different aggregates and non-interacting molecules in NFA thin films. Then, we will discuss in detail the EA spectral characteristics and the extracted dipole moment ($\Delta\mu$) and polarizability ($\Delta p$) of the NFAs with different loading ratios and at different energies of excitation. The EA results will be further correlated with the results coming from DFT calculations. Finally, a four-state model considering the LE and CT excitations will be exploited to elucidate the modifications in $\Delta\mu$ and $\Delta p$ upon Y6 aggregation. This work provides in-depth insight into the structure-property-performance correlation in Y6 non-fullerene acceptors, which will aid the design and development of high-performance organic photovoltaic cells.

## Results

### Optical absorption of Y6 and ITIC in solution and thin films

The UV-vis absorption spectra of Y6 and ITIC in solution (20–25 mg/ml) and thin films are shown in Fig. 1. The absorption spectrum of the Y6 thin film is more redshifted and broadened as compared to that of ITIC. In Y6, when going from solution to thin film, a significant redshift (around 180 meV) of the maximum peak intensity is observed. On the other hand, in ITIC, the corresponding redshift is only around 80 meV. The red shift in the absorption spectrum observed upon aggregation was originally rationalized by Kasha and further discussed by Spano; it is usually attributed to a combined effect of intermolecular electronic couplings, vibronic couplings, and intermolecular charge transfer (formation of J-type and H-type aggregates)[27–29].

To further investigate the effect of NFA's intermolecular interactions on the absorption spectral characteristics, we fabricated thin films with different loading ratios of the NFA molecules in an insulting polymer matrix. Polyvinylcarbazole (PVK) was chosen as the polymer matrix for its extensive miscibility with NFAs and its large bandgap over 3.40 eV, which forms a 'Type-I heterojunction' with the NFAs and leads to negligible CT between the two materials. As demonstrated in Supplementary Fig. 2, PVK can well disperse the Y6 and ITIC molecules in spin-coated thin films, whereas Y6 tends to aggregate in other commonly used insulating polymers, such as PMMA (polymethyl methacrylate) and PS (polystyrene). This solid-solvation method is widely used to study aggregation-induced emission and quenching processes in organic-light-emitting diode (OLED) materials[46]. We implemented the same method here to investigate the excitonic

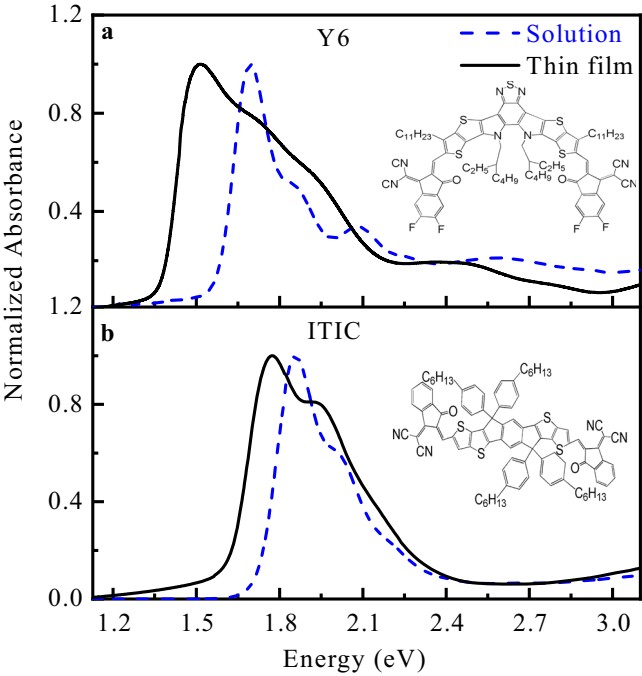

**Fig. 1 | UV-vis absorption spectra of solution and thin films of Y6 and ITIC.**
Normalized UV-vis absorption spectra of **a** Y6 and **b** ITIC dissolved in chloroform
($CHCl_3$) solution (blue dashed line) and thin films (solid black line) spin-coated on a
quartz substrate. The concentration of solution used is 20–25 mg/ml. The chemical
structures of Y6 and ITIC are shown in the inset of the graphs.

properties of single molecules and aggregated molecules of Y6 and
ITIC in thin films. Figure 2a, b shows the thin-film UV-vis absorption
spectra with different loading ratios of Y6 and ITIC coated on quartz
substrates, respectively. The detailed changes in the spectral char-
acteristics are summarized in Fig. 2c, d. For small loading ratios, 1 wt%
to 10 wt% (weight%) of NFAs in PVK, the spectral line shape and peak
positions resemble the spectrum in the corresponding solution with
only a small redshift of 0.02–0.05 eV. This indicates that the NFA
molecules are well isolated and dispersed in the polymer matrix with
weak inter-molecular electronic coupling. In the case of Y6, noticeable
redshift and broadening of the absorption spectra are observed when
the loading ratio is larger than 10 wt%. More importantly, the redshift
and broadening of the Y6 thin films continue to increase as the loading
ratio increases. This could be attributed to the stabilization of CT
excitation energies and increase in inter-molecular electronic cou-
plings as a result of a decrease in inter-molecular distances due to
tighter packing. In the case of ITIC, both the absorption spectral
characteristics and redshift only show little changes even at high
loading ratios, suggesting that there is only weak short-range elec-
tronic coupling among ITIC molecules and lack of CT contribution to
the excited states of ITIC molecular aggregates, as is confirmed by our
DFT calculations (see Supplementary Figs. 3 and 4(a, b) in the Sup-
plementary Information). Furthermore, it has been pointed out that its
banana-type molecular conformations facilitate the formation of long-
range aggregation networks in Y6[47,48].

## Molecular packing in spin-coated thin films

The evolution of molecular packing in Y6 with different loading ratios
in the solid-solution thin films is also confirmed by the
grazing–incidence wide-angle X-ray scattering (GIWAXS) results
(Supplementary Fig. 5). The lamellar peaks in Y6 become noticeable
when the loading ratio is higher than 10 wt% (Supplementary Fig. 5g),
and the π−π diffraction peak starts to emerge when the loading ratio is
larger than 40% (Supplementary Fig. 5h). We intended to probe the

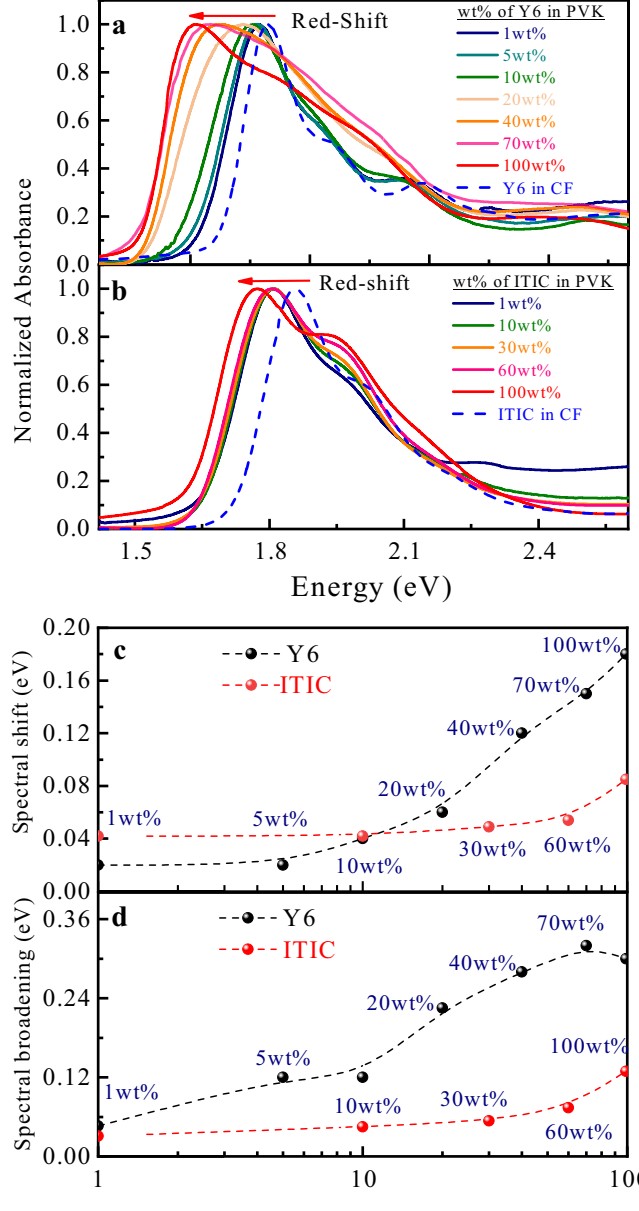

**Fig. 2 | UV-vis absorption spectra of thin films having Y6 and ITIC dispersed
in PVK.** UV-vis absorption spectra of solid-state thin films having different loading
ratios (1–100 wt%) of **a** Y6 and **b** ITIC in a PVK matrix. **c** Spectral shift (eV) and
**d** spectral broadening (eV) for different concentrations of Y6 (red symbols with
dashed line) and ITIC (black symbols with dashed line) relative to those in solution.
The spectral shift is defined as the energy difference of the maximum absorption
intensity between thin films and solution. The spectral broadening is calculated by
finding the difference in spectral full-width half maximum (FWHM) between thin
films and solution.

evolution of different aggregate contributions with increasing Y6
concentrations in PVK. Due to weak scattering signals for the low
loading ratios of Y6 in PVK, we were not able to identify the detailed
packing motif of Y6. On the other hand, it is consistent with the optical
absorption results indicating that there are no Y6 aggregates formed in
loading ratios of 10 wt% or lower. In addition, from the out-of-plane
signals (Supplementary Fig. 5h), it is confirmed that Y6 shows face-on
orientation (around $q_z = 1.8$ Å$^{-1}$) from 40 wt% (very low) to 70 wt% Y6 in
PVK, which becomes even stronger in the pristine Y6 thin film. Such
preferred orientation in a pristine Y6 thin film is also observed in the
first harmonic analysis of the EA spectra, as discussed in the

Supplementary Information (see Supplementary Note 2). Consistently, in these high-loading ratios, we have observed the largest changes in optical absorption. To further confirm the packing configurations, we also tested GISAXS under synchrotron radiation and observed that the amorphous phase size is decreased when the loading ratio of Y6 is higher than 10 wt%, as shown in Supplementary Table 1 in Supplementary Information.

We further complemented these findings by spectrally decomposing the optical absorption spectra into different aggregates and non-interacting regions using a Frank−Condon-weighted density of states (FCWD) in the framework of the Marcus−Levich−Jortner theory[49], as shown in Supplementary Fig. 6. This fitting analysis provides detailed information about the non-interacting and interacting Y6 molecules at different energy excitations and loading ratios in PVK, see Supplementary Table 2 that collects all the Franck−Condon fitting parameters for Y6. We recall that:

$$FCWD = \frac{1}{\sqrt{4\pi\lambda k_B}} \sum_{n=0}^{\infty} \exp(-S)\frac{S^n}{n!}exp\left[-\frac{(\Delta E + n\hbar\omega + \lambda)^2}{4\lambda k_B}\right] \quad (3)$$

where $\lambda$ denotes the Marcus reorganization energy (meV); S, the Huang−Rhys factor accounting for the vibrational coupling; $k_B$, Boltzmann constant; and $\Delta E$, the energy difference between the energy of the considered vibrational peak and the 0-0 transition energy $E_{00}$ (eV).

The absorption spectrum of the Y6 solution can be well reproduced by two states ($S_1$ and $S_2$) and their progressions with $E_{00}$ energies at 1.69 eV and 2.10 eV, respectively. For fitting, we used a vibrational energy $\hbar\omega_1$ equal to 160 meV for the first FC progression and optimized the peak intensity values (A), Huang−Rhys parameter (S) and classical reorganization energy parameter ($\lambda$) within a reasonable range of values for the different transitions. Upon going to the solid state and having aggregates appearing, it should be noted that inter-molecular electronic coupling will lead to energy splitting and the appearance of levels at lower energies (denoted as LowEn) and higher energies (denoted as HighEn) with respect to the $E_{00}$ values for isolated molecules. Moreover, since each progression represents the electronic configurations of isolated molecules and various aggregates, the FC parameters should be similar in different samples. Therefore, during the fitting, the values of the FC parameters for each progression were kept almost the same for all samples; only the peak intensity values were adjusted to get the best spectral fit. Importantly, the results obtained in this way are consistent with the values reported by Köhler and co-workers using a similar approach[50]. We note that these authors reported the formation of two types of aggregates/dimers, which they referred to as Agg. I and Agg. II, see further discussion below.

When the loading ratio of Y6 in PVK is increased to 10 wt%, the contribution from Agg. II starts to emerge in the absorption spectrum along with the contribution from the non-interacting molecules. For the neat Y6 film, besides the contribution from Agg. II and non-interacting Y6 molecules, there is also a significant contribution from Agg. I. Both these aggregates have low-energy (solid line) and high-energy (dashed line) components, with Agg. I at 1.42 eV/1.79 eV and Agg. II at 1.51 eV/1.70 eV. The increased contribution of Agg. I around the $S_1$ state in neat Y6 confirms our findings of stronger diffraction peaks observed in the GIWAXS data (Supplementary Fig. 5). Moreover, this gives an insight into the emergence of different aggregates and their contribution to the spectral changes in the optical absorption with an increased loading ratio of Y6 in thin films. A detailed investigation of the effect of these different aggregates, as determined from Franck−Condon progressions, on the charge transfer properties of Y6 at different excitation energies will be discussed below.

## Electroabsorption analysis on Y6 and ITIC

We sought to investigate more quantitatively the effect of molecular packing on the changes in electronic properties in NFAs by EA spectroscopy. Considering the Stark effect as described in Eq. (2), due to the change in energy of a state $\Delta E$ under an electrical field, the change in optical transmittance of an organic film under a sinusoidal electrical field perturbation can be expressed as:

$$\left(\frac{\Delta T}{T}\right)_{2\omega} = \left[\frac{1}{4}(\Delta p)\frac{\partial A_D}{\partial E} + \frac{1}{12}(\Delta\mu^2)\frac{\partial^2 A_D}{\partial E^2}\right]\frac{V_{ac}^2}{0.43d^2}\sin\left[2\omega t + \frac{\pi}{2}\right] \quad (4)$$

where $\Delta p$ denotes the difference in polarizability (cm³) and $\Delta\mu$ is the difference in dipole moment (Debye) between the ground and excited states. $V_{ac}\sin(\omega t)$ is the applied AC voltage (V) with an amplitude $V_{ac}$ and an angular frequency $\omega$, and $d$ denotes the organic film thickness (nm). $\left(\frac{\Delta T}{T}\right)_{2\omega}$ is the change in device optical transmittance with respect to the second harmonic of the modulating AC signal. The second harmonic signal $\left(\frac{\Delta T}{T}\right)_{2\omega}$ is considered first as it is independent of the static dipole effect due to directional molecular packing that could complicate the analysis. $\frac{\partial A_D}{\partial E}$ and $\frac{\partial^2 A_D}{\partial E^2}$ are the first and second derivatives of the device absorbance ($A_D = 0.43\alpha d$, where α is the absorption coefficient and d, the thickness respectively). In this work, EA$_{2\omega}$ was measured in transmission (T) mode and analyzed using the device absorbance data. We had previously demonstrated that this approach could effectively eliminate the influence of the strong optical interference and electro-refraction effects that are usually inherent in thin-film devices[51]. Details of the derivation of Eq. (4) are described in the Supplementary Nnote 1. According to Eq. (4), it can be readily seen that a resemblance of the measured EA$_{2\omega}$ spectrum to the first derivative of the device absorbance $\frac{\partial A_D}{\partial E}$ indicates a dominant contribution from $\Delta p$, whereas a resemblance to the second derivative $\frac{\partial^2 A_D}{\partial E^2}$ indicates a dominant contribution from $\Delta\mu$. As discussed above, it is generally expected that an excited state with strong CT character should have a large increase in exciton radius, and its EA$_{2\omega}$ spectral characteristics would be mainly contributed by $\Delta\mu$[42–45].

Figure 3 shows the EA$_{2\omega}$ spectra of Y6 with different loading ratios wt% (1%, 10%, 40%, 70%, and 100%) in PVK, where the devices have a general structure of ITO (140 nm)/NFA: PVK (150-250 nm)/Al(15 nm). The spectral characteristics of all devices can be generally divided into three regions. We first consider Region I, around 1.30−1.70 eV, corresponding to the $S_0 \rightarrow S_1$ transition. The excitonic properties of this transition play a significant role in the OPV processes, as both the directly excited and relaxed excitons from higher excited states will come down to that state. For 1 wt% and 10 wt% Y6 samples, although the EA$_{2\omega}$ spectral characteristics are mainly dominated by the first derivative, there is a minor contribution from the second derivative of the corresponding absorbance spectra. This points out that there is some extent of CT character present in Y6 single molecules. Upon increasing the loading ratios from 40 wt% to 70 wt%, there is an obvious increase in the contribution from the second derivative, and the spectrum becomes purely second derivative in the 100% pristine Y6 sample. This suggests that the strong CT character in neat Y6 thin film is largely contributed by molecular aggregations. The EA$_{2\omega}$ results are consistent with the UV-vis results in that the redshift and broadening of the absorption spectra continue to increase with increasing loading ratios. It should be noted that, as shown in Supplementary Fig. 6, Region I in the 100 wt% Y6 consists of Agg. I and Agg. II LowEn contributions, while Region I in 10 wt% Y6 only consists of Agg. II LowEn. Therefore, the much stronger CT character in the 100 wt% Y6 suggests that the CT character in Agg. I LowEn is much stronger than in Agg. II LowEn. More quantitative differentiation on the CT character between the two aggregates might further bring insight into the contribution from different packing configurations. However, due to the large degree of overlap between different progressions, a more

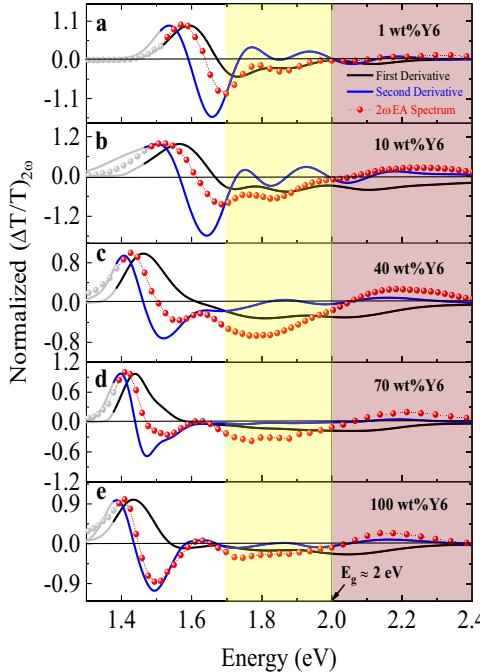

**Fig. 3 | Second harmonic EA spectra at different loading ratios of Y6 in PVK.**
**a–e** The red symbols with dotted lines refer to the measured second harmonic (2ω) EA signals (transmission mode) at different loading ratios (1 wt%, 10 wt%, 40 wt%, 70 wt% and 100 wt%) of Y6 in PVK polymer. The black and blue solid lines denote the first and second derivatives of device absorbance, respectively. The first excited state ($S_1$ state) is denoted by Region I, where the gray symbols and solid lines at low energy indicate the uncertainty region where absorbance has low signal-to-noise ratio. The dispersed phase (non-interacting molecules) of Y6 molecules is defined by Region II, which is shaded in yellow. Region III represents the transport band gap ($E_g$) of Y6, which is approximately 2 eV. The applied electric field is around $10^5$ V/cm.

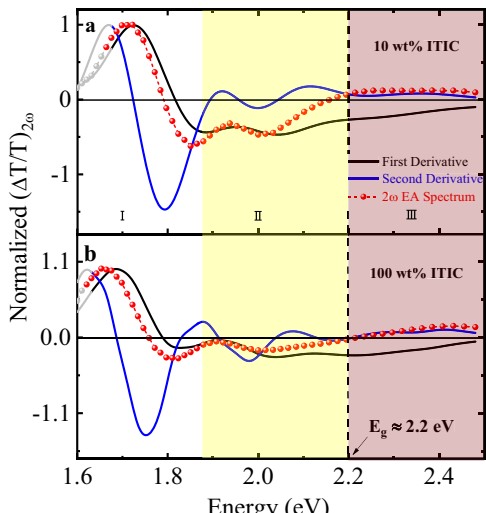

**Fig. 4 | Second harmonic EA spectra of different loading ratios of ITIC in PVK.**
The red symbols with dotted line refer to the measured second harmonic (2ω) EA signals (transmission mode) of **a** 10 wt% ITIC in PVK and **b** 100 wt% ITIC. The black and blue solid lines denote the first and second derivatives of device absorbance, respectively. The first excited state ($S_1$ state) is denoted by Region I, where the gray symbols and solid lines at low energy indicate the uncertainty region where absorbance has low signal-to-noise ratio. The dispersed phase (non-interacting molecules) of ITIC molecules is defined by Region II, which is shaded in yellow. Region III represents the transport band gap ($E_g$) of ITIC, which is approximately 2.2 eV. The applied electric field is around $10^5$ V/cm.

careful study to disentangle the individual contribution through optical simulation is required.

In Region II, around 1.70–2.00 eV, the $EA_{2\omega}$ spectral characteristics for all the devices mainly resemble the first derivative of the corresponding absorption spectra, independent of the loading ratios. It suggests that the excitons generated in this region have negligible CT character, even at high loading ratios. As depicted in Supplementary Fig. 6, it is found that the absorption in this region is mainly contributed by the non-interacting Y6 molecules. Kupgan et al. also reported that around 30% of the Y6 molecules do not form aggregates in solution-processed Y6 thin films[52]. Similar results were also reported recently by Köhler and co-workers in their study mentioned above of different aggregates in Y6 and N4 NFAs[50]. Furthermore, the $EA_{2\omega}$ spectral characteristics of all samples in Region III above 2.0 eV resemble the second derivative of the corresponding absorption spectra, indicating excitons generated in this region have a strong CT character. Recent photoemission spectroscopy data and computational results have determined the transport gap energy ($E_g$) in Y6 to be 2.0 eV[17,53]. This explains the strong CT character observed in $EA_{2\omega}$ in this region. It should be noted that the transport gap energy determined in the above $EA_{2\omega}$ result is independent of the loading ratios. This indicates that the intermolecular interactions have a strong influence only on the lower excitonic $S_1$ state but a small impact on the charge-transport states due to their higher energy and degenerate character. Considering the energy onset of the $S_1$ state from the absorption spectrum in solution, the $S_1$ state exciton binding energy in Y6 molecules is 400 meV. However, it is clear that using the same approach to determine the binding energy in a neat Y6 film is invalid, as the $S_1$ state there already has a strong CT character that could facilitate free charge generation[26].

As a comparison, $EA_{2\omega}$ measurements were also conducted on ITIC with different loading ratios in PVK. Figure 4 shows the EA spectra of 10 wt% and 100 wt% ITIC in PVK devices. In contrast to Y6, the EA spectra are similar for both low- and high-loading ratio devices, and the spectral characteristics of the first excitonic transition $S_0 \rightarrow S_1$ around 1.6 eV to 1.9 eV mostly resemble the first derivative of the corresponding device absorbance. This result is consistent with the small changes observed in the absorption spectra of the ITIC thin films with increasing the loading ratios. This result, as will be discussed below, can be attributed to the negligible contribution of the CT excitation to the excited states in ITIC aggregates. Consequently, the $S_1$ in ITIC has mainly a LE (Frenkel) character. On the other hand, like Y6, there is a transition from the first to the second derivative at around 2.2 eV corresponding to the transport gap energy, which is also consistent with the previously reported values determined by photoemission spectroscopy[53]. The $S_1$ state exciton binding energy in ITIC molecules is determined to be around 500-600 meV.

We next fitted the first optical transition in the $EA_{2\omega}$ spectra of devices with different loading ratios of Y6, where the energy range can be simply approximated by a Gaussian analysis. Figure 5 shows the fitting results around the first excitonic transition $S_0 \rightarrow S_1$ using Eq. 4. Details of the fitting procedures can be found in Supplementary Note 3 in the Supplementary Information. Surprisingly, the $\Delta\mu$ value in pure Y6 film (8.35 D) is similar to, or even smaller than that in the 1 wt% and 10 wt% Y6 film (-9.5 D). In contrast, the $\Delta p$ value reduces by a factor of 3 upon increasing the loading ratio, from 290 Å³ in 1 wt% Y6 device to 95 Å³ in the pure Y6 device. In the case of ITIC, both diluted and pure films have similar $\Delta p$ and $\Delta\mu$ values, consistent with the observations described above.

## Correlation between the CT character of the excited states and the $EA_{2\omega}$ data

In order to rationalize the $EA_{2\omega}$ data and to quantify the CT character of the Y6 excitations, we performed excited-state calculations using molecular dimers extracted from the crystal structures[14,15]. An interesting point that was largely overlooked in previous discussions of Y6-

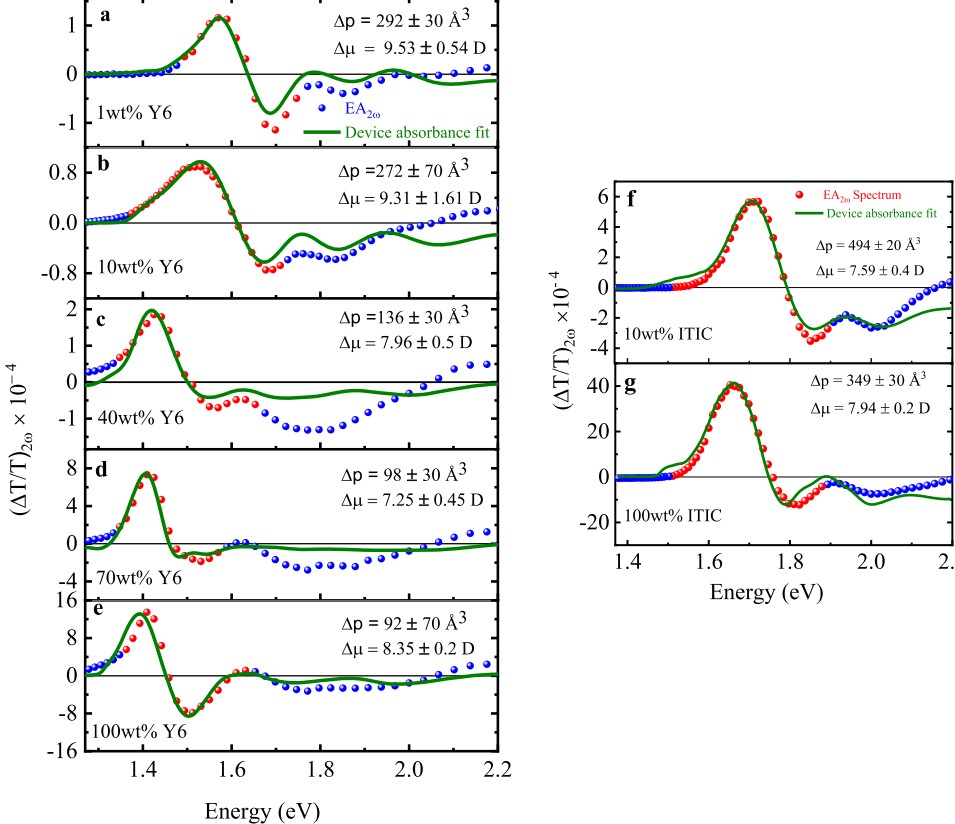

**Fig. 5 | Electroabsorption spectra fitting. a–e** Measured second harmonic (2ω) EA signals (transmission mode) for different loading ratios (1 wt%, 10 wt%, 40 wt%, 70 wt%, and 100 wt%) of Y6 in PVK fitted with Eq. (4). **f, g** Second harmonic (2ω) EA signals of 10 and 100 wt% ITIC in PVK fitted with Eq. (4). The red symbols represent the fitted region of the EA signal, which corresponds to the first Gaussian band of the thin film absorbance of Y6 and ITIC. The extracted Δµ (change in dipole moment) and Δp (change in polarizability) values of the different samples are shown in the figures.

aggregate excited states comes from the recognition that in Y6 crystals there are two sets of monomers that differ by their conformations. The monomer geometry of the first class ($M_1$) shows a twisted (helical) conformation while the molecules in the second class ($M_2$) have a bent conformation with the two terminal acceptor groups pointing in the same direction (see Fig. 6). The energy of the $S_1$ states in bent and twisted monomers differ by about 300 meV. The DFT calculations show that the average Δµ value related to Y6 $S_0 \rightarrow S_1$ excitation is about 8 D, which is in good comparison with the EA data. This relatively large Δµ value can be attributed to an intra-molecular CT character of the $S_0 \rightarrow S_1$ excitation in Y6 monomers (see Fig. 6). The average Δp value in the Y6 monomers is estimated to be about 800 Å³, which is somewhat larger than the experimental value.

As a result of the existence of two classes of monomers, all dimers can also be classified into two groups, *i.e.*, symmetric and asymmetric. One might speculate that the symmetric and asymmetric sets of dimers might correspond to the two types of aggregates deduced from the fitting of the optical spectra. The DFT calculations show that the $S_0 \rightarrow S_1$ excitation in all dimers has a significant inter-molecular CT character. However, the CT contribution is particularly large in asymmetric dimers containing one $M_1$ monomer and one $M_2$ monomer. Figure 6 displays the NTOs of an asymmetric dimer where the CT and LE contributions are nearly equal (dimer A). However, asymmetric dimers with nearly 100% CT contribution to $S_1$ exist as well (dimer C), see Supplementary Figs. 7 and 8 and Supplementary Table 3. Moreover, the DFT results indicate that, in dimers with pure CT excitations, the Δp value is very small. Finally, we note that in ITIC crystals, the DFT calculations underline that the dimers have a pure local-exciton (*i.e.*, intra-molecular) character (see Supplementary Figs. 3 and 4 and Supplementary Table 4).

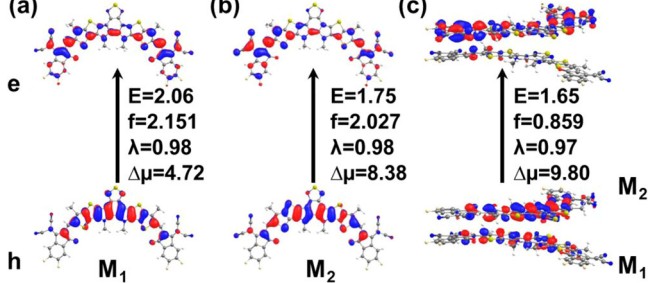

**Fig. 6 | Types of monomers and dimers formed in Y6 crystal.** Natural transition orbitals (NTOs) of the (**a**) twisted ($M1$) and (**b**) bent ($M2$) monomers and (**c**) the dimer formed from these two monomers. Here, E denotes the transition energy in eV, f represents the oscillator strength, λ is the eigenvalue of the NTO, and Δµ signifies the dipole moment change in Debye for the $S_0 \rightarrow S_1$ transition.

In previous reports, the strong CT character observed in some organic systems has been usually associated with the large Δµ values determined by EA[41–43]. However, it was also shown that, in general, the Δµ does not correlate well with the CT weight[54]. This is expected to be especially problematic for Y6, as Δµ originates from both intra-molecular and inter-molecular CT transitions ($\Delta\mu_{intra}$ and $\Delta\mu_{inter}$). We note that, in amorphous Y6 films, there will be a distribution of $\Delta\mu_{intra}$ and $\Delta\mu_{inter}$ parameters, so there will be cases when $\Delta\mu_{intra}$ and $\Delta\mu_{inter}$ can cancel each other or act in a concerted manner.

In order to shed more light on this issue, we considered a four-state model involving the two LE and two CT excitations of a dimer as discussed above. This model was successfully used earlier to rationalize the EA properties in organic molecular dimers[55,56]. Based on the

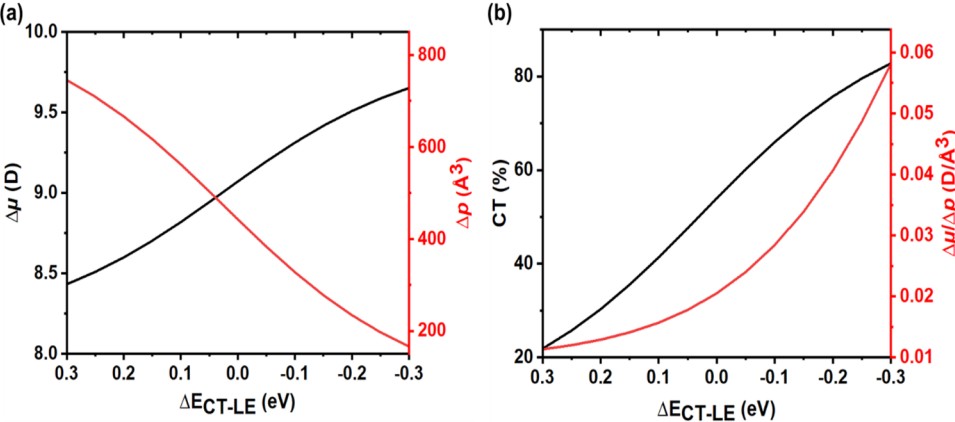

**Fig. 7 | Correlation of Δμ and Δp with charge transfer weight (CT%) of the dimer $S_1$ excitation, as calculated from the four-state model. a** Variations in Δμ and Δp values of Y6 dimers (black and red solid lines, respectively) as a function of the energy difference between the LE and CT states ($\Delta E_{CT-LE}$). **b** Comparison of the ratio of Δμ/Δp (red solid line) to the charge transfer weight (black solid line) vs. the energy separation between the LE and CT states ($\Delta E_{CT-LE}$).

DFT results, we considered that the interaction of pure Y6 intermolecular CT excitations with the electric field is due to the $\Delta\mu_{inter}$ coupling, while both $\Delta\mu_{intra}$ and $\Delta p_{intra}$ interaction mechanisms are involved in the case of pure LE excitations (see Supplementary Note 4). We also assumed that, as the Y6 loading ratios in Y6: PVK increase, the CT state energy is stabilized due to a decrease in inter-molecular distances. Using the results generated by the four-state model, we computed the dependence of Δμ and Δp for a Y6 dimer as a function of energy separation between the LE and CT states ($\Delta E_{LE-CT}$); the results are shown in Fig. 7. We note that, for the sake of simplicity, we kept fixed the values of the other microscopic parameters although they are also rigorously distance dependent (see Supplementary Note 4). As seen from Fig. 7a, as the CT state energy stabilizes ($\Delta E_{CT-LE}$ decreases), Δμ increases while Δp decreases; as a consequence, $\Delta\mu/\Delta p$ increases as well. The calculations also show (Fig. 7b) that, as $\Delta E_{CT-LE}$ decreases, the CT contribution to the $S_1$ state (CT%; this parameter is given by the square of the $c_{CT}$ coefficient from Eq. (1)) continuously increases as expected. As the energy of the CT excitation is located below that of the LE excitation ($\Delta E_{CT-LE} < 0$), a sharp increase in the $\Delta\mu/\Delta p$ value is also observed with the increase in CT% parameter.

Interestingly, a similar trend for $\Delta\mu/\Delta p$ is also obtained for Y6 from the derived EA experimental data. Indeed, as shown in Fig. 8, the $\Delta\mu/\Delta p$ ratio increases by a factor of 3 from 0.03 D/Å³ in 1 wt% and 10 wt% Y6 to 0.09 D/Å³ in pure Y6. Moreover, the increase in $\Delta\mu/\Delta p$ ratio correlates with the changes in the optical absorption spectra. Therefore, we conclude that the increase in CT character (CT%) of the Y6 excitations that take place during aggregate formation is the main reason for the observed red shift of the absorption band and the trends shown by the Δμ and Δp values. On the other hand, the $\Delta\mu/\Delta p$ ratio in the case of ITIC remains the same, whether a diluted ITIC: PVK film or a neat ITIC film is considered. This is again consistent with the trend shown by the absorption band and the DFT results indicating that the excited states in ITIC films have a dominant LE (Frenkel) nature.

## Contribution of different aggregates on CT property

To further demonstrate the contribution of different aggregates on the CT character, we attempted to fit the $EA_{2\omega}$ spectra of 10 wt% and 100 wt% Y6 at different energies of excitation. The result can also be used to verify the correlation between the CT character and the $\Delta\mu/\Delta p$ value. First, we tried to decompose the 10 wt% and 100 wt% Y6 $EA_{2\omega}$ spectra into different regions based on the aggregated and non-interacting molecules obtained from the Franck−Condon progression analysis (Supplementary Fig. 9). In the case of 10 wt% Y6, the Y6 molecules are mostly isolated with little CT character around the first

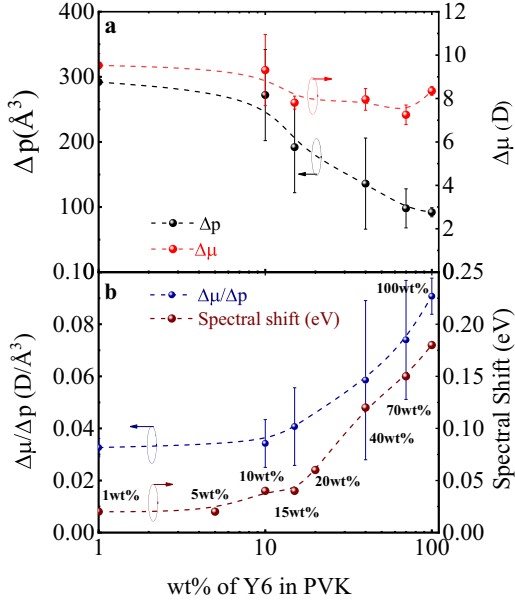

**Fig. 8 | Summary of Δμ and Δp at different loading ratios of Y6 in PVK. a** Δμ and Δp values extracted from the fitting of the second harmonic EA spectra at different loading ratios of Y6 in PVK. **b** Comparison of the ratio of Δμ/ Δp (blue dashed line) to the UV-vis absorption spectral shift (brown dashed line) from Fig. 2c at different loading ratios of Y6 in PVK.

excitonic transition. As shown in Supplementary Fig. 9a, Region 1, which corresponds to the $S_1$ state (1.37−1.65 eV), is mainly contributed by the low-energy part of Agg. II (Agg. II, LowEn); Region 2 (1.72−1.93 eV) has more contribution from the non-interacting molecules, whereas Region 3 (above 2.03 eV) has contribution from the high-energy part of Agg. II (Agg. II, HighEn), which is also above the transport energy level (~2 eV)[17,53]. Table 1 summarizes the EA fitting results. Region 2 has the smallest $\Delta\mu/\Delta p$ value, indicating a weak CT character. Region 1 has a slightly higher $\Delta\mu/\Delta p$ value, pointing to an increased CT character. In Region 3, the $\Delta\mu/\Delta p$ value is ten times higher than that in Region 2, suggesting a very strong CT character. In this region, Δμ is extremely large (26.02 ± 3 Debye) and Δp is significantly reduced; this is possibly the case because the excitations in Region 3 fall into the transport energy levels with a high probability of free charge generation.

**Table 1 | Summary of Δμ and Δp values extracted from EA fitting, their ratio, and nature of excitons at different aggregated and non-interacting regions for 10 wt% and 100 wt% Y6**

| Loading ratio of Y6 | Aggregated/ non-interacting regions | | Energy (eV) | Δμ (D) | Δp (Å³) | Δμ/Δp (D/Å³) | FE/ CT |
|---|---|---|---|---|---|---|---|
| 10 wt% | Region 1 | Agg. II, LowEn | 1.37–1.65 | 9.31 ± 1.2 | 292 ± 50 | 0.032 ± 0.01 | Little CT |
| | Region 2 | Non-interacting | 1.72–1.93 | 10.18 ± 0.5 | 577 ± 40 | 0.018 ± 0.01 | FE |
| | Region 3 | Agg. II, HighEn | 2.03–2.41 | 26.02 ± 3 | 94 ± 70 | 0.277 ± 0.19 | CT |
| 100 wt% | Region 1 | Agg. I&II, LowEn | 1.3–1.64 | 8.35 ± 0.2 | 92 ± 10 | 0.091 ± 0.01 | CT |
| | Region 2 | Non-interacting | 1.64–1.87 | 6.95 ± 0.8 | 349 ± 30 | 0.020 ± 0.01 | FE |
| | Region 3 | Agg. I&II, HighEn | 1.97–2.41 | 13.61 ± 2 | 44 ± 30 | 0.311 ± 0.19 | CT |

In the case of 100 wt% Y6, the film consists of various aggregate/ dimer configurations and exhibits strong CT character at the first excitonic excitation. As shown in Supplementary Fig. 9b, Region 1 (1.31–1.64 eV) represents the low-energy excitations of aggregates I and II (Agg. I & II, LowEn); Region 2 (1.64–1.87 eV) represents the non-interacting molecules of Y6, whereas Region 3 (above 1.97 eV) represents the high-energy excitations of the aggregates (Agg. I & II, HighEn) as well as the transport energy level. According to the EA fitting results shown in Table 1, Region 2 has the smallest $\Delta\mu/\Delta p$ ratio, consistent with the value obtained in the 10 wt% sample. Region 1 has a much larger $\Delta\mu/\Delta p$ value, indicating its strong CT character. It should be noted that Region 1 in the 100 wt% Y6 sample consists of Agg. I & II LowEn, while Region 1 in 10 wt% sample only consists of Agg II LowEn. Therefore, the much higher (3 times) $\Delta\mu/\Delta p$ value in the 100 wt% Y6 sample suggests that the CT character in Agg. I, LowEn is much stronger than that in Agg. II, LowEn. Similar to the 10 wt% sample, Region 3 in the 100 wt% Y6 sample also has a very high $\Delta\mu/\Delta p$ value, which is possibly due to its strong CT character stemming from being above the transport energy level. Although the above analysis still has limitations in terms of separating the contributions from different aggregates due to their considerable spectral overlap, the results provide a clear differentiation of the contributions to the CT character from aggregated and non-interacting Y6 molecules. More importantly, the EA analysis at different energies demonstrates the validity of the proposed $\Delta\mu/\Delta p$ ratio for the assessment of the CT character.

## Discussion

In summary, we have reported a comprehensive investigation of the effect of molecular packing on the charge transfer characteristics in the Y6 and ITIC NFAs via electroabsorption (EA) spectroscopy. We found that, in the case of Y6, both molecular and aggregate excitations are characterized by a charge-transfer (CT) character (intra-molecular and inter-molecular, respectively). We have demonstrated that the intra- and inter-molecular (CT) excitations can be decoupled by fabricating solid-solution thin films with different loading ratios of the NFAs in a non-interacting polymer matrix. According to the EA data, as a result of an intra-molecular CT nature, the $S_0 \rightarrow S_1$ transition in isolated Y6 molecules is characterized by a large intra-molecular dipole moment change ($\Delta\mu_{intra}$). Molecular packing leads to the hybridization of intra-molecular excitations with inter-molecular CT excitations that are naturally characterized by larger $\Delta\mu_{inter}$ values. As a result of the fact that both intra and inter-molecular excitations have significant dipole moment values, the overall dipole moment after aggregation does not vary substantially. However, as confirmed by both density functional theory (DFT) calculations and the use of a four-state model, the existence of inter-molecular CT excitations results in small $\Delta p$ values due to the energetic stabilization of the CT states; consequently, the hybridization of LE (Frenkel) excitations with intermolecular CT excitations leads to a substantial decrease in the $\Delta p$ value upon aggregation. In contrast, in ITIC, due to the absence of CT contributions, both molecular and aggregated excited states have a purely Frenkel excitation nature. Our work has thus brought deep insight into the correlation between inter-molecular charge transfer and molecular electronic configurations.

## Methods

### Device fabrication

The non-fullerene acceptors Y6 and ITIC were purchased from Tin Hang Technology Limited. The ITO patterned glass substrates were cleaned by sequential ultra-sonication in acetone, alcohol, and dried using a high-purity nitrogen gun. Ultraviolet (UV) ozone surface treatment was performed at room temperature for 10–15 min. Y6 and ITIC were spin-coated from a chloroform solution (20–25 mg/ml) to form thin films of 100–200 nm thickness inside a nitrogen-filled glovebox. No annealing was performed for Y6 and ITIC films. For the dispersed Y6 and ITIC films, a different loading ratio (1 wt%, 10 wt%, 40 wt%, and 70 wt%) of NFA was added to the PVK (polyvinyl carbazole) polymer solution to prepare the required concentration. The samples were then deposited with 15 nm aluminum as the top electrode through a shadow mask using vacuum thermal evaporation process at a base pressure around $10^{-7}$ to $10^{-6}$ Torr. The semi-transparent devices were fabricated for electroabsorption measurement in transmission mode.

### EA spectroscopy

EA spectroscopy (Transmission mode) was conducted to measure the $EA_{1\omega}$ and $EA_{2\omega}$ signals. The setup is equipped with a light source (Xenon Arc Lamp 1000 W, Newport), monochromator (Zolix), optical chopper (Thorlabs), calibrated silicon and germanium photodetectors (Thorlabs), low-noise current pre-amplifier (Stanford Research Systems, SR570), lock-in amplifier (Stanford Research Systems, SR830), and a function generator (SRS DS360). A monochromatic beam is transmitted through the semi-transparent device (not encapsulated) and detected by using the silicon/germanium photodetectors. During the measurement, the samples were housed inside a vacuum cryostat (Oxford Instruments) at base pressure around $10^{-5}$–$10^{-6}$ Torr. While measuring the device transmittance (T), the optical chopper provides a synchronous reference signal (190 Hz) to the lock-in amplifier. The transmitted light intensity (T) was detected by silicon photodetector, which generated a current signal and fed it into the lock-in amplifier. This measured transmittance (T) was used to calculate the derivatives. To measure the electric field-induced change in transmittance (ΔT), a function generator was used to modulate the internal electric field in the organic layer by superimposing a sinusoidal voltage at a frequency of 1 kHz on a negative DC voltage. The applied electric field is around $10^5$ V/cm. The modulated signal from the detector was amplified using the current preamplifier by choosing a suitable gain or sensitivity. The lock-in amplifier was connected to demodulate the signal, phase referenced to the function generator at the different harmonics of the modulation fundamental frequency. The harmonic number in the lock-in amplifier can be adjusted to measure the first and second harmonic EA signals. The measured ΔT needs to be scaled by a factor of √2 to convert the root-mean-square value (RMS) to the peak value.

## Optical absorbance and film thickness

Thin film (spin-coated on quartz substrates) and solution absorbance were measured using a Perkin-Elmer Lambda 1050 + UV/Vis/NIR spectrophotometer, and the corresponding film thickness was measured by Bruker OM-Dektak profilometer.

## GIWAXS

Cu X-ray source (8.05 keV, 1.54 Å), Pilatus3R 300 K detector, and Xeuss 2.0 SAXS/WAXS laboratory beamline were used to perform the GIWAXS and GISAXS measurements. The incidence angle is 0.2°. In this experiment, the grazing-incident X-ray beam size at sample stage, divergence, and sampling area were $400 \times 60\ \mu m^2$ (W×H), $80 \times 30\ \mu$ rad (*W×H*), and ~1.15 mm². The sample-to-detector distance was chosen to allow SAXS studies to detect momentum transfers [$q = 4\pi \sin\theta/\lambda$, where $2\theta$ is the scattering angle] of SAXS experiments was $0.004–0.18\ Å^{-1}$.

## Electronic-structure calculations

The electronic structure calculations of the ground and excited states of Y6 and ITIC monomers and dimers were performed at the density functional theory (DFT) level and its time-dependent variant (TD-DFT). For these calculations, we used the long-range corrected ωB97X-D functional and 6–31 G (d, p) basis set, where the range-separation parameter (ω) was set at $0.0126\ Bohr^{-1}$, which is taken from our previous report[52]. We considered the implicit dielectric environment based on the polarizable continuum model (PCM) with a dielectric (ε) value of 3.0, which is commonly used for organic solar cells. All the DFT calculations were performed with the Gaussian 16 package[57].

## Data availability

All data supporting the results of this study are available in the paper and the Supplementary Information. Additional data related to this work are available from the corresponding authors upon request.

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

## Acknowledgements

The authors acknowledge the financial support from the General Research Fund (CityU 11303618 and CityU 11317422) from the Research Grants Council of Hong Kong SAR, China. The work at the University of Arizona was funded by the UA College of Science and the Office of Naval Research, Award No. N00014-20-1-2110 and No. N00014-24-1-2114.

## Author contributions

S.M., T.L., and S.-W.T. conceived the idea. H.Y.H. and S.O. conducted the UV-vis experiment. S.M. fabricated the devices and conducted the electroabsorption measurement and analysis. S.M. P., J.L.B., and V.C. performed and analyzed the electronic-structure calculations. Y.L. and X.L. conducted and analyzed the GIWAXS and GISAXS measurements. S.M., T.L., S.M.P., J.L.B., V. C., and S.-W.T. contributed to the manuscript preparation. P.C.Y.C., H.L.Y., and S.K.S. provided valuable contributions to the result discussion and manuscript preparation. All the authors discussed the results and commented on the manuscript.

## Competing interests

The authors declare no competing interests.
