## [Peer Review File · Nature Communications]

Assessing Intra- and Inter-Molecular Charge Transfer
Excitations in Non-Fullerene Acceptors Using
Electroabsorption SpectroscopyREVIEWER COMMENTS

Reviewer #3 (Remarks to the Author):

In the work, the authors address the excited state properties of Y6 and ITIC in different states (solution, neat solid films, solid solutions with PVK) using optical absorption and electroabsorption (EA) spectroscopy. EA is a powerful tool to investigate the nature of molecular excited states. The study reveals important differences between the two NFAs. Importantly, while ITIC exhibits rather little changes in the optical and EA spectra upon increasing the NFA molecular concentration in PVK, Y6 shows a significant (and continuous) redshift of the absorption accompanied by a characteristic change of in the EA spectra. Notably, while the EA 2ω of the low concentration sample of Y6 is dominated by the first derivative of the absorption, the spectrum becomes fully dominated by the second derivative for the neat Y6 film. The data are interpreted in terms of an increased CT character with increasing domain size, counterbalanced by a reduced wavefunction delocalization. The authors also perform EA experiments in the presence of a DC electric field, which for the Y6 solid films indicates orientational anisotropy, as expected.

The manuscript provides interesting experimental inside. However, the present version falls short of being suited for publication in Nat Communication for two reasons.

First, the analysis of the data remains at a rather qualitative level (while the title suggests a “Quantitative Analysis of Intermolecular Charge Transfer”). I acknowledge that the EA spectra are properly analyzed with well-established equations, but the interpretation of the values of $\Delta\mu$ and Δp with the proposed exciton model (Figure 6) remains at a very qualitatively level. If I understand correctly, the transition from a first derivative-type EA to a second-derivative-type EA with increasing Y6 concentration comes from a decrease of Δp while $\Delta\mu$ is fairly independent of concentration. How does this correlate quantitatively with the characteristic parameters of the exciton model (charge separation distance and wavefunction delocalization)? Does charge separation distance increase with Y6 concentration or is the main effect a reduction of charge delocalization? And are such conclusions quantitatively support by theory? Related to this, I had problems to understand the claim in the abstract “Moreover, it is found that the conservation of exciton polarizability plays a key role in the strong CT character in Y6, instead of the large increase in dipole moment.” What does this really mean? I thought that the exciton polarizability (Δp) changes with concentration. The same criticism holds for the analysis of the EA measurements in presence of a DC field. Again, the data and fits are good but the conclusions are qualitative: “The increasing non-zero crossing spectral characteristics while increasing the loading ratio in Y6 devices strongly support that there is a preferred molecular orientation in Y6 thin film” I would have a more quantitative analysis of the spectra in terms of the orientational distribution of Y6.

My second criticism concern the potential overlap of this paper with Ref. 26. I have currently no access to Ref 26 (Manuscript under revision NENERGY- 22030456) but the text in this manuscript suggests that it's also about the CT character of excitons in aggregated molecules of Y6 and its derivatives. So, what is different here? The methods, the samples? Without having access to that reference, its not possible to assess the novelty of the current submission.

I finally noticed the "Results" part of the manuscript is rather short (less than seven manuscript pages) and what is named "Discussion" is already the conclusion. So, there is plenty of space for a comprehensive analysis and interpretation of the data.

Reviewer #4 (Remarks to the Author):

Comments to the authors:

This manuscript mainly reports a quantitative investigation on the effect of molecular packing on the charge transfer characteristics in Y6 and ITIC. The author found that Y6 has strong charge transfer (CT) character due to molecular stacking, and the origin of this CT character by the conservation of polarizability from the ground state to the excited state. In addition, the author also proposed a model based on the overlapping of the electron and hole wavefunction to elucidate this special mechanism. Overall, the work is well organized and reveals some interesting results. Thus, I would like to recommend it to be published in Nature Communication after some revisions. Here are some questions need to be addressed in a revised manuscript:

(1) There are some written mistakes in this manuscript and further corrections and rechecks are needed. For example, it is recommended to modify "A-D-A '- D-A" to "A-DA'D-A" on page 3, line 57. There should be spaces between numbers and units in the manuscript. The format of references 16 and 36 is not consistent with other references. The full name of DFT should be given when it first appears on page 3, line 63. The chemical structures of Y6 and ITIC in Figure 1 need to be redrawn to clearly display the single and double bonds in the molecules.

(2) The author studied the relationship between molecular stacking and charge transfer characteristics. Has the author considered the impact of different solvent treatments on the molecular stacking of thin films?

(3) On page 8, the author cited multiple stacking modes of Y6 obtained through MD simulation. In fact, single crystal data of Y6 and ITIC have been reported so far. Therefore, the authors are suggested to use molecular stacking modes in single crystal structures to be more accurate.

(4) The optimal molecular conformation of Y6 is C-type, which is significantly different from ITIC (S-type), resulting in different molecular stacking modes. In order to demonstrate the role of conformation in determining the stacking mode, it is recommended that the author choose the ultra-narrow bandgap acceptor DTPC-DFIC (DOI: 10.1021/jacs.7b13239; 10.1002/adfm.202204255) with a C-type conformation as a comparative explanation.

Reviewer #5 (Remarks to the Author):

Mahadevan and his colleagues conducted a study on the charge transfer properties of two model non-fullerene acceptors (NFAs), namely Y6 and ITIC, by dispersing NFA molecules in a PVK polymer with different loading ratios and employing electroabsorption spectroscopy. According to their findings, the authors argue that the molecular packing of Y6 can give rise to strong charge transfer characteristics. Additionally, the results indicate that the strong charge transfer behavior in Y6 is primarily influenced by the preservation of polarizability rather than a significant increase in dipole moment. The experimental and theoretical data confirm the presence of strong charge-transfer characteristics in Y6 aggregates. However, similar conclusion has been extensively debated in prior research utilizing experimental and theoretical approaches (e.g., ref. 47, 15, or *Angew. Chem., Int. Ed.* 2021, 60, 15348, & etc.). Therefore, it is crucial to justify the novelty beyond available works before considering its publication in *Nature Communications*. I have provided detailed comments below.

- 1) In NFAs adopting the A-D-A or A-D-A'-D-A architecture, the strong intramolecular push-pulling effect induces the formation of excitations characterized by charge transfer (CT) properties. Additionally, previous studies on Y6 films have demonstrated the delocalization of excitons, the formation of intra-moiety states with CT characteristics, or even the direct separation into free charges. Hence, the detection of CT characteristics in Y6 aggregates through electroabsorption (EA) spectroscopy is not unexpected. However, the manuscript fails to address the crucial aspect of how excitations with CT properties influence the subsequent charge generation process.
- 2) In Figure 3, it seems the major characters of EA signals are dominant in Region I. What is the reason for this?
- 3) The authors propose that the absorption in Region II is primarily contributed by non-interacting Y6 molecules, even in the neat Y6 film. While this may hold true for Y6 solutions or films with low loading ratios, it raises the question of why non-interacting Y6 molecules are still present in the 100% pristine Y6 sample. Further clarification or evidence regarding the existence of non-interacting Y6 molecules in the fully pristine sample is necessary.
- 4) The determination of fitting parameters in Figure S3 and Table 2 appears to be somewhat coarse. It is crucial to experimentally or theoretically determine the transition energies and vibrational frequencies in Y6 solutions and films. Obtaining such data would significantly enhance the accuracy and reliability of the fitting parameters.
- 5) When an external field is applied, both the ground state and excited state energies can be altered. Distinguishing the contributions from the ground state and excited states to the measured EA signals is important. The charge transfer characteristics of excited states may indeed play a more significant role in the photon-to-current conversion process. Exploring and discussing the impact of both ground and excited states on the EA signals would provide a comprehensive understanding of the system.
- 6) In a film sample, multiple configurations of Y6 aggregations can exist, each characterized by different optical responses. It is crucial to consider this effect when interpreting the EA data. Accounting for the presence of different aggregation configurations and their respective optical responses would contribute to a more accurate and comprehensive analysis of the experimental results.

Response to referees

Many thanks to the reviewers' insightful and constructive comments; those are very helpful for us in improving the quality of our work. Accordingly, we have extensively revised the manuscript with additional discussion, analysis, and experimental results. All the changes have been highlighted in the revised manuscript, and the detailed replies to the comments are as follows. We hope the reviewer will find our responses satisfactory.

Reviewer #3:

The manuscript provides interesting experimental inside. However, the present version falls short of being suited for publication in Nat Communication for two reasons.

(1) First, the analysis of the data remains at a rather qualitative level (while the title suggests a "Quantitative Analysis of Intermolecular Charge Transfer"). I acknowledge that the EA spectra are properly analysed with well-established equations, but the interpretation of the values of $\Delta\mu$ and Δp with the proposed exciton model (Figure 6) remains at a very qualitatively level. If I understand correctly, the transition from a first derivative-type EA to a second derivative-type EA with increasing Y6 concentration comes from a decrease of Δp while $\Delta\mu$ is fairly independent of concentration. How does this correlate quantitatively with the characteristic parameters of the exciton model (charge separation distance and wavefunction delocalization)? Does charge separation distance increase with Y6 concentration or is the main effect a reduction of charge delocalization? And are such conclusions quantitatively support by theory?

Response:

Thanks a lot for your constructive comment and kind effort in reviewing our manuscript. Regarding the correlation between the characteristic parameters extracted by EA and the exciton models, we agree with the reviewer that it is an important direction. Still, there are some fundamental issues that need to be considered. As the reviewer might also notice, the existing

theories describing excitons are mainly based on a hydrogenic model with electrons orbiting the holes inside a molecule. However, in an actual situation, the electron might not simply orbit around the holes, especially when the electron and hole orbitals are in different moieties within a molecule. In the case of a dimer, the excited electrons and holes could even be localized in different molecules. Therefore, the hydrogenic exciton model might not truly reflect the actual situation, where the estimated characteristic parameters (e.g., exciton radius and binding energy) can only provide a qualitative estimation. Similarly, the hydrogenic model was also considered in the Stark effect; the extracted Δp and $\Delta\mu$ values from EA might not quantitatively reflect the actual value of the characteristic parameters. However, according to the Stark effect, the change in absorption (energy) of an excited state has different dependences (**Appendix I** in Supplementary Information) on the polarizability Δp and dipole moment $\Delta\mu$. These two parameters can bring insight into the electronic properties of the excited states. It is particularly reliable in comparing the changes of the two parameters in similar material systems, as demonstrated in this work with different loading ratios of Y6 in PVK. (The above discussion has been included in the revised manuscript as highlighted on **page 12**).

According to our EA results, the increase in charge-transfer characteristic with increasing the Y6 concentration is attributed to the reduced delocalization compared to that in the Y6 monomer. Exciton dissociation is dependent on the escape probability of electrons from the electrostatically attracted holes. Generally, a larger separation distance (dipole moment) between electron and hole would facilitate the exciton dissociation, as previously observed in some material systems¹⁻³. However, charge delocalization should also play a role in exciton dissociation, as illustrated in **Figure 6**. Given the same charge separation distance, a smaller delocalization could reduce the chance of orbitals overlapping, which eventually increases the probability of dissociation. Consequently, an efficient exciton dissociation should have a large $\Delta\mu$ and a small Δp . By comparing different loading ratios of Y6 and at different energies of excitation, we have found that the ratio $\Delta\mu/\Delta p$ can better describe the charge transfer character of the excited states. (The above discussion has been modified to improve the readability in the revised manuscript as highlighted on **page 13**).

Instead of comparing with the analytical models, the above EA results and the proposed model are supported by the density-functional-theory (DFT) calculation results. The table below shows the extracted Δp and $\Delta\mu$ values from EA fitting, and the overlapping parameter (O_{e-h}) and fluorescent oscillator strength (f_{flu}) calculated by DFT as shown in Ref [54] in the main text⁴.

Y6	DFT calculation		EA Fitting	
	Overlapping parameter (O_{e-h})	Oscillator strength (f_{flu})	Δp ($\times 10^{-22} cm^3$)	$\Delta\mu$ (Debye)
Monomer	0.65	2.0	2.72 ± 0.7	9.31 ± 1.6
Dimer 1, T-T	0.60	1.4		
Dimer 2, CC-TT	0.40	0.2	0.92 ± 0.1	8.35 ± 0.2
Dimer 3, CT-CT	0.58	0.7		

Table 1. $\Delta\mu$ and Δp values extracted from EA fitting compared to the overlapping parameter (O_{e-h}) and fluorescent oscillator strength (f_{flu}) calculated by DFT.

As discussed in the manuscript (page 13, line 354), the polarizability of a state is correlated to the electron/hole wavefunction delocalization. In the case of the Y6 monomer, the overlapping between electron and hole wavefunctions (O_{e-h}) and oscillator strength (f_{flu}) are larger when compared to those in dimers. At the same time, Δp extracted by EA shows a larger value ($2.72 \pm 0.7 \times 10^{-22} cm^3$). In the case of Y6 dimers, all configurations show smaller overlapping parameters and oscillator strength than those in the monomer. In particular, the CC-TT dimer has the smallest O_{e-h} and f_{flu} , which has previously been reported with the strongest electronic coupling between molecules and therefore the charge transfer characteristics^{5,6}. Consistently, the extracted Δp in pure Y6 film is three times smaller than that found in isolated molecules. Thus,

the change in polarizability extracted by EA is correlated to the overlapping parameter determined by DFT, and consistent with the proposed model of the contribution of Δp on exciton dissociation.

(2) Related to this, I had problems to understand the claim in the abstract “Moreover, it is found that the conservation of exciton polarizability plays a key role in the strong CT character in Y6, instead of the large increase in dipole moment.” What does this really mean? I thought that the exciton polarizability (Δp) changes with concentration.

Response:

Thanks for your comment. “Conservation of exciton polarizability” refers to a small Δp value, which means that the excited state polarizability is nearly the same as that in the ground state. In previous EA studies, only the $\Delta\mu$ (difference in permanent dipole moment) is associated with the charge transfer (CT) character¹⁻³. However, we have found that the pristine Y6 film which has the strongest CT character shows nearly the same $\Delta\mu$ value as that in the dispersed Y6 thin films. On the other hand, Δp in the pristine Y6 film is much smaller. Indicating that the increased CT character in the pristine Y6 film is attributed to the reduced electron/hole orbitals delocalization as illustrated in **Figure 6**. For the same amount of increase in charge separation ($\Delta\mu$), the reduced delocalization (Δp) reduces the probability of the electron/hole wavefunction overlapping. This increases the chance for exciton separation. Therefore, the contribution of Δp which has been overlooked in previous studies, should play a key role in determining the CT character.

To further confirm the contribution of Δp on the CT character, in the revised manuscript, additional results on the investigation of the excitonic properties at different excitation energies are included. Taking 10 wt% and 100 wt% Y6 as the examples, as shown in **Supplementary Fig. 4**, we first identified the regions of (1) aggregates, (2) non-interacting molecules, and (3) transporting gap by Frank-Condon weighted density of states (FCWD) analysis. After that, the $\Delta\mu$ and Δp values of each region were extracted by the EA analysis.

In the case of 10 wt% Y6, where the Y6 molecules mostly remain isolated with little CT character around the E_{S1} transition. The part-by-part fitting curves are shown in **Figure R1**, and corresponding results are summarized in **Table R1**. The extracted $\Delta\mu$ and Δp values in Region 1 and Region 2 are similar, resulting in similar $\Delta\mu/\Delta p$ values (1.7 – 3.2). Suggesting the weak CT character in aggregate II. On the other hand, the $\Delta\mu/\Delta p$ value in Region 3 is one order of magnitude higher, contributed by the increased $\Delta\mu$ and reduced Δp values. It is expected, as Region 3 falls into the excitation above the transporting gap (2.0 eV)⁷.

Figure R1. Electroabsorption spectra fitting. Measured second harmonic (2ω) EA signals (transmission mode) of 10 wt% of Y6 in PVK fitted with Equation 2. The red symbols represent the fitted region of the EA signal, which corresponds to the (a) low energy aggregate (Agg. II, LE), (b) non-interacting molecules, and (c) high energy aggregate (Agg. II, HE), regions of thin

film absorbance of 10 wt% Y6. The extracted $\Delta\mu$ (change in dipole moment) and Δp (change in polarizability) values of different aggregate and non-aggregated configurations are shown in the figure.

FCWD (10 wt% Y6)		Energy (eV)	$\Delta\mu$ (D)	Δp (10^{-22} cm^3)	$\Delta\mu/\Delta p$ $\times 10^{22} \text{ D}$ $/\text{cm}^3$	FE/ CT
Region 1	Agg. II, LE	1.37-1.65	9.31 ± 1.2	2.92 ± 0.5	3.19 ± 0.5	Little CT
Region 2	Non- interacting	1.72-1.93	10.18 ± 0.5	5.77 ± 0.4	1.76 ± 0.1	FE
Region 3	Agg. II, HE	2.03-2.41	26.02 ± 3.2	0.94 ± 0.7	27.68 ± 18.5	CT

Table R2. $\Delta\mu$ and Δp values extracted from EA fitting, their ratio and nature of excitons at different aggregated and non-interacted regions of 10 wt% Y6 in PVK.

In the case of 100 wt% Y6, where the film consists of various dimer configurations has exhibited strong CT character at the E_{S1} excitation. The part-by-part fitting curves are shown in **Figure R2** and corresponding results are summarized in **Table R3**. The extracted $\Delta\mu$ values are similar in both Regions 1 and 2, but Δp in Region 1 is three times smaller which results in a larger $\Delta\mu/\Delta p$ value. Interestingly, compared to Region 1 in the 10 wt% Y6, Region 1 in 100 wt% Y6 has significantly increased contribution from the aggregate I, suggesting the different CT characters in different aggregates, and it is consistent with the recent theoretical results^{5,6}. Further detailed analysis may even be possible to identify the CT contributions from different dimer configurations in different absorption bands, but it is out of the scope of this study. Similarly, Region 3 in 100 wt% Y6 also has a very high $\Delta\mu/\Delta p$ value as observed in 10 wt% Y6, owing to the strong CT character in the above transporting gap excitation.

Figure R2. Electroabsorption spectra fitting. Measured second harmonic (2ω) EA signals (transmission mode) of 100 wt% of Y6 fitted with Equation 2. The red symbols represent the fitted region of the EA signal, which corresponds to the low energy aggregate (Agg. I & II, LE), non-interacting molecules, and high energy aggregate (Agg., I & II HE), regions of thin film absorbance of pure Y6. The extracted $\Delta\mu$ (change in dipole moment) and Δp (change in polarizability) values of different samples are shown in the figure. The table below shows the extracted $\Delta\mu$ and Δp values from EA fitting, their ratio, and the nature of excitons at different bands of 100 wt% Y6.

FCWD (100 wt% Y6)	Energy range (eV)	$\Delta\mu$ (D)	Δp (10^{-22} cm^3)	$\Delta\mu/\Delta p$ $\times 10^{22} \text{ D}$ $/\text{cm}^3$	FE/ CT	
Region 1	Agg. I & II, LE	1.31 – 1.64	8.35 ± 0.2	0.92 ± 0.1	9.07 ± 0.3	CT
Region 2	Non- interacting	1.64- 1.87	6.95 ± 0.8	3.49 ± 0.3	1.99 ± 0.2	FE
Region 3	Agg. I & II, HE	1.97- 2.41	13.61 ± 2.2	0.44 ± 0.3	30.93 ± 18.9	CT

Table R3. $\Delta\mu$ and Δp values extracted from EA fitting, their ratio and nature of excitons at different aggregated and non-interacted regions of 100 wt% Y6.

In summary, the above results of EA analysis at different excitation energies further confirm the roles of Δp and $\Delta\mu/\Delta p$ (as comparing Region 1 and 2 in both cases) on accounting for the CT character of an excited state.

(3) The same criticism holds for the analysis of the EA measurements in presence of a DC field. Again, the data and fits are good but the conclusions are qualitative: “The increasing non-zero crossing spectral characteristics while increasing the loading ratio in Y6 devices strongly support that there is a preferred molecular orientation in Y6 thin film” I would have a more quantitative analysis of the spectra in terms of the orientational distribution of Y6.

Response:

Thanks for your comment. We agree with the reviewer that the current analysis is qualitative. To our best knowledge, there is still no reported quantitative analysis of the dipole effect from the non-zero crossing in the EA spectrum^{8,9}. To obtain more quantitative results corresponding to the orientation distribution as suggested by the reviewer, we believe that another set of systematic EA measurement and analysis approaches considering the orientations of both

incident light and electrical field should be considered. In addition, it will also require mathematical models to handle the statistical distribution of the molecular orientation in film. It is definitely a very good direction to be further developed for the EA technique, as molecular orientation has been playing significant roles in various organic electronic devices. Although the present results in our works are qualitative, the observed increasing non-zero crossing spectral characteristics provide strong evidence of the evolution from highly dispersed to preferred molecular orientation in films while increasing the Y6 concentration. It is consistent with the GISAXS results (**Supplementary Fig. 3**), and it provides strong support for the understanding of the changes in CT character in different concentrations.

According to the reviewer's comment, we have revised our title to "On the Study of Intra- and Inter-Molecular Charge Transfer in Non-Fullerene Acceptors by Electroabsorption Spectroscopy" to better reflect the scope of our works.

(4) My second criticism concern the potential overlap of this paper with Ref. 26. I have currently no access to Ref 26 (Manuscript under revision NENERGY- 22030456) but the text in this manuscript suggests that it's also about the CT character of excitons in aggregated molecules of Y6 and its derivatives. So, what is different here? The methods, the samples? Without having access to that reference, it's not possible to assess the novelty of the current submission.

Response:

Thanks for your comment. Regarding the concerns about the overlapping of the two papers, we have already communicated with the editor. Here, we also attach the preprint version of the manuscript in the Reference section for your reference. The NENERGY manuscript investigates the long exciton lifetime and reduced non-radiative recombination rate of Y-series acceptors⁴. Experimentally, it mainly relied on photoluminescence approaches, transient PL, and PLQY. The experimental results are compared with the non-adiabatic coupling in different materials calculated by DFT. The NENERGY manuscript has the EA data of a dispersed and a pristine non-

fullerene acceptor film in the Supporting Information to support the stronger CT character in the pristine Y6 film.

In terms of the methodologies, the present work presents a more comprehensive EA measurement and analysis of the CT properties from isolated to aggregated Y6 and ITIC molecules in thin films. Including detailed analysis of the contributions from different aggregates on the optical absorption spectra by the FCWD approach. Here, we cite the DFT calculation results in the NENERGY manuscript, as the calculated overlapping parameter of the electron-hole wavefunctions and the oscillator strength has a strong correlation with the polarizability changes extracted from the EA analysis.

In terms of the findings, the NENERGY manuscript has provided a more holistic picture in view of the degree of non-adiabatic coupling to illustrate the strong CT character in Y6. On the other hand, the present work is more focused on the discussion of the evolution of the electronic properties ($\Delta\mu$ and Δp) extracted by EA analysis and their correlation to molecular aggregation and corresponding CT mechanism.

(5) I finally noticed the “Results” part of the manuscript is rather short (less than seven manuscript pages) and what is named “Discussion” is already the conclusion. So, there is plenty of space for a comprehensive analysis and interpretation of the data.

Response:

Thanks for your suggestion. We have revised the manuscript with additional discussion on EA results of different loading ratios of Y6 in PVK, fitting analysis, exciton recombination model and additional EA analysis on different energies of excitation, the length of “Results” in the revised manuscript has been extended to 11 pages.

Reviewer #4 (Remarks to the Author):

(1) There are some written mistakes in this manuscript and further corrections and rechecks are needed. For example, it is recommended to modify “A-D-A ' - D-A” to “A-DA'D-A” on page 3, line 57. There should be spaces between numbers and units in the manuscript. The format of references 16 and 36 is not consistent with other references. The full name of DFT should be given when it first appears on page 3, line 63. The chemical structures of Y6 and ITIC in Figure 1 need to be redrawn to clearly display the single and double bonds in the molecules.

Response:

Thanks for your constructive comment. We have made the necessary corrections in the revised manuscript as per your suggestions.

(2) The author studied the relationship between molecular stacking and charge transfer characteristics. Has the author considered the impact of different solvent treatments on the molecular stacking of thin films?

Response:

Thanks for your comment. We agree with the reviewer that different solvent treatments would affect molecular stacking. In this work, we are aiming to use a solvent that can allow us to fabricate well-dispersed (isolated) Y6 in thin film and increase its molecular packing while increasing the Y6 concentration. Among those solvents, we have found that chloroform is an ideal candidate, due to the excellent solubility of Y6 and PVK in chloroform. Besides the solvent effect, we also tried different solvent additives (e.g., DIO and CN) and insulating polymers (e.g. PS and PMMA), as shown in **Figures R3 and R4** respectively. Surprisingly, only PVK can well disperse the Y6 molecules, both PS and PMMA will induce Y6 aggregation even at a very low concentration. Especially in the case of Y6 in PS, as shown in **Figure R5**, only 10 wt% of Y6 in PS already resembles

the absorption spectrum of pure Y6. The corresponding EA spectrum (Figure R6) also shows a strong contribution from the second derivative of the absorption spectrum. Those works on the effect of solvent, solvent additives, and polymer matrix are still ongoing, hopefully, we can share it in the near future.

Figure R3. The normalized absorption spectrum of Y6 with different concentrations (0.5%, 5%) of solvent additive, DIO. The spectral red-shift and narrowing of the absorption spectrum indicate increased molecular aggregation induced by DIO.

Figure R4. The normalized absorption spectra of different concentrations (1 wt%, 5 wt%, and 100 wt%) of Y6 dispersed in different polymer matrices such as Polystyrene (PS), Polymethyl

methacrylate (PMMA), Polyvinyl Carbazole (PVK). It is evident that only PVK can disperse Y6 in thin films.

Figure R5. The normalized absorption spectrum of different concentrations (10 wt%, 50 wt%) of Y6 dispersed in Polystyrene (PS) polymer.

Figure R6. The electroabsorption spectrum of 10wt% of Y6 dispersed in Polystyrene (PS) polymer. It exhibits a strong second derivative contribution at its S1 state, with CT excitons.

(3) On page 8, the author cited multiple stacking modes of Y6 obtained through MD simulation. In fact, single crystal data of Y6 and ITIC have been reported so far. Therefore, the authors suggested using molecular stacking modes in single-crystal structures to be more accurate.

Response:

Thanks for your comment. The reason why choosing the MD simulation results is that the simulation was intended to replicate the molecular packing in solution processed Y6 thin films. It is more relevant to our films which were prepared by spin-coating. In addition, Yip and co-workers have found that the molecular stacking modes in Y6 single crystal also appear in films prepared by spin-coating⁵.

(4) The optimal molecular conformation of Y6 is C-type, which is significantly different from ITIC (S-type), resulting in different molecular stacking modes. To demonstrate the role of conformation in determining the stacking mode, it is recommended that the author choose the ultra-narrow bandgap acceptor DTPC-DFIC (DOI: 10.1021/jacs.7b13239; 10.1002/adfm.202204255) with a C-type conformation as a comparative explanation.

Response:

Thank you for the insightful suggestion. We did a literature study on this ultra-narrow bandgap acceptor DTPC-DFIC and compared its conformation with Y6 and ITIC. As you have mentioned, Y6 shows C-type molecular conformation which helps the molecule to form dimers through terminal and core interactions. In addition, the planar structure of Y6 and DTPC-DFIC is beneficial to orbital overlapping, electron delocalization, and bandgap reduction due to the intramolecular charge transfer (ICT) effect^{10,11}. Additional discussion on the contribution of the conformation of

Y6 on molecular packing has been included in the revised manuscript with the suggested references from the reviewer.

Reviewer #5 (Remarks to the Author):

(1) Similar conclusion has been extensively debated in prior research utilizing experimental and theoretical approaches (e.g., ref. 47, 15, or *Angew. Chem., Int. Ed.* 2021, 60, 15348, & etc.). Therefore, it is crucial to justify the novelty beyond available works before considering its publication in *Nature Communications*. I have provided detailed comments below.

Response:

Thanks for the constructive comment. As the reviewer has mentioned, molecular packing facilitated charge transfer (CT) has been extensively debated. Theoretically, it has been demonstrated that the conformation of the Y acceptor and its different dimer configurations contribute to the strong CT character^{5,6}. Experimentally, several transient absorption spectroscopy (TAS) studies have demonstrated the generation of CT excitons upon photoexcitation^{12,13}. Considerable free-charge generation has also been observed in Y6 homojunction photovoltaic (PV) device structure¹⁴. Despite the success of theoretical studies, the reported TAS and PV experimental results are still phenomenological. It is still unclear how the observed charge generation and CT excitons related to the different electronic properties in Y6. Here below are the highlights of the novelty beyond those previously reported studies.

In terms of the methodologies, using electroabsorption (EA) spectroscopy allows us to gain deeper insight into the correlation between the electronic properties and the CT properties in Y6. The polarizability (Δp) and dipole moment ($\Delta \mu$) of the excited state extracted by EA analysis are correlated to the delocalization and separation of the excitons. In addition, using the solid-solution approach, for the first time, we were able to reveal the evolution of the above two parameters from isolated molecules to increasing molecular aggregation. Combining EA results with the Franck-Condon weight density (FCWD) analysis and Gaussian deconvolution on optical absorption spectra, allows us to identify the excitonic properties in a single molecule and aggregate Y6 at different energies of excitation.

In terms of the findings, besides the general wisdom that the aggregated Y6 has strong CT character, surprisingly, we have found that the isolated Y6 molecules already have little CT character (combination of the first and second derivative features in 1 wt% Y6 thin film), indicating the conformation of the Y6 molecule would also contribute to the CT character. Moreover, instead of the commonly expected contribution from the large increase in dipole moment (electron-hole separation), we have found that the increased CT character observed in pure Y6 film is attributed to the reduced delocalization of the electron/hole wavefunctions. As shown in **Figure 7** in the manuscript, the isolated and aggregated Y6 films have similar dipole moment, but the aggregated film has largely reduced polarizability. Consequently, the proposed model derived from the EA results as shown in **Figure 6** has excellent agreement with overlapping parameters in Y6 monomer and dimers obtained by the density functional theory (DFT) as shown in **Table R1**.

Therefore, our works not only experimentally demonstrate approaches (solid-solution thin films, EA, and FCWD analysis) to correlate the electronic properties ($\Delta\mu$ and Δp) and CT characters in single molecule and aggregated Y6, but also reveal the important role of charge delocalization on the CT process in Y6. It brings insight into the future molecular design to allocate the electron/hole orbitals more favourable for efficient charge generation.

(2) In NFAs adopting the A-D-A or A-D-A'-D-A architecture, the strong intramolecular push-pulling effect induces the formation of excitations characterized by charge transfer (CT) properties. Additionally, previous studies on Y6 films have demonstrated the delocalization of excitons, the formation of intra-moiety states with CT characteristics, or even the direct separation into free charges. Hence, the detection of CT characteristics in Y6 aggregates through electroabsorption (EA) spectroscopy is not unexpected. However, the manuscript fails to address the crucial aspect of how excitations with CT properties influence the subsequent charge-generation process.

Response:

We agree with the reviewer that the CT character in Y6 aggregates is expected, and it has been commonly ascribed to the strong electron coupling in its core-to-core (CC) molecular packing configuration. However, it has been recently demonstrated that a PM6:Y6 OPV device with mainly terminal-to-terminal (TT) packing can also achieve high efficiency¹⁵. Therefore, it is still questionable whether the strong CT character observed in Y6 is solely due to the molecular packing or it is inherent in the molecular structure. In this study, we have indeed observed certain degree of CT character already exists in isolated Y6 molecules (second derivative features in 1 wt% Y6 thin film). As for comparison, it is purely Frenkel-type excitation in ITIC, as its EA spectrum perfectly resembles the first derivative features. More importantly, we have also found that the further enhanced CT character in aggregated in Y6 is not contributed by the increase in $\Delta\mu$, but due to a reduced Δp value. Different from the previous reports that the increased CT character is often associated with the increase in $\Delta\mu$ ¹⁻³. Since Δp is correlated to the delocalization of the electron/hole wavefunction, we propose that the smaller Δp in aggregated Y6 film eventually reduces the overlapping of the electron and hole wavefunctions. This is consistent with the DFT calculation results that the overlapping parameter between the electron and hole wavefunctions (**Table R1**) in Y6 dimers is smaller than that in the monomer. The reduced overlapping increases the probability of exciton dissociation which brings insight into the mechanism of the recently observed free charge generation in Y6 thin films.

(3) In Figure 3, it seems the major characters of EA signals are dominant in Region I. What is the reason for this?

Response:

Thanks for the question. The EA signal in organic materials is usually stronger at the first excitation level as in Region I. As depicted in Equation (2) in the manuscript, the EA signal is directly proportional to the derivative of the optical absorption ($\partial A/\partial E$). Since most organic materials have steeper optical onset (larger $\partial A/\partial E$) at the first excitation state, the EA signals

are usually dominant in Region I. As an example of exception as shown in **Figure R7**, PCBM has a very gentle increase of absorption at the absorption onset, the EA signal around that region is smaller in amplitude.

Figure R7. Measured absorption and second harmonic (2ω) EA signal (transmission mode) of pristine $PC_{71}BM$ thin film.

(4) The authors propose that the absorption in Region II is primarily contributed by non-interacting Y6 molecules, even in the neat Y6 film. While this may hold true for Y6 solutions or films with low loading ratios, it raises the question of why non-interacting Y6 molecules are still

present in the 100% pristine Y6 sample. Further clarification or evidence regarding non-interacting Y6 molecules in the fully pristine sample is necessary.

Response:

Thanks for the comment. It is believed that not all the molecules in the film can form dimers or aggregates, especially using the solution fabrication process. Kupgen et al. recently conducted molecular dynamic (MD) simulations to mimic the solution-processed Y6 thin film and calculate the population of different Y6 dimers and non-interacting molecules in solid thin film⁶. Authors have found that 45% of the molecules form dimers in the TT configuration, and 25% of the molecules form CC-TT and CT-CT configurations, leaving 30% of the Y6 molecules to remain non-interacting in the film. Furthermore, through the Franck-Condon analysis as shown in **Supplementary Fig. 4** in the Supplementary Information, we have found that there is still some contribution from the non-interacting Y6 molecules to the optical absorption in the 100% pristine Y6 sample. In this revised manuscript, we have included the EA analysis on different excitation energies in pristine Y6 samples, it is confirmed that Region 2 indeed shows a weaker CT character (small $\Delta\mu/\Delta p$), indicating the presence of non-interacting molecules.

(5) The determination of fitting parameters in Figure S3 and Table 2 appears to be somewhat coarse. It is crucial to experimentally or theoretically determine the transition energies and vibrational frequencies in Y6 solutions and films. Obtaining such data would significantly enhance the accuracy and reliability of the fitting parameter.

Response:

Thanks for the constructive comment. We agree with the reviewer comment that determining the Franck Condon (FC) fitting parameters experimentally or theoretically can significantly enhance the accuracy of the analysis. In fact, our results are inspired by the work reported by

Kroh et al¹⁶. They have performed a detailed Franck-Condon analysis to deduce the formation of different aggregate and non-aggregate chromophores of both Y6 and another derivative N4 in solutions and neat films. They first fitted the optical absorption of the Y6 solution, where the spectrum predominantly contained non-interacting molecules. It can be well reproduced by two FC progressions with transition energies determined by the peak local maxima at around 1.69 eV (E_{01}) and 2.1 eV (E_{02}). The vibrational energy, $\hbar\omega_1 = 160$ meV, they used was based on the Raman measurements of Y6¹⁷. The peak intensity values (A), Huang-Rhys parameter (S), and Gaussian linewidth parameter (λ) were adjusted within a reasonable range of values for different transitions to obtain a good fit. The fitting equation of the absorption spectra with the Frank-Condon-weight density of states (FCWD) in the framework of Marcus-Levich-Jortner theory¹⁸ is expressed as;

$$FCWD = \frac{1}{\sqrt{4\pi\lambda k_B}} \sum_{n=0}^{\infty} \exp(-S) \frac{S^n}{n!} \exp\left[-\frac{(\Delta E + n\hbar\omega + \lambda)^2}{4\lambda k_B}\right]$$

where λ is the Gaussian linewidth parameter (50 to 100 meV), S is the Huang-Rhys factor accounting for the coupling of the two states (0 to 1), k_B is the Boltzmann constant, and ΔE is the energy difference between the oscillation energy and E_{00} (eV).

In this manuscript, we followed the same fitting approach to analyze the aggregate contributions in thin films with different loading ratios of Y6 in PVK polymer. As shown in **Figure 2**, the measured absorption spectra of the Y6 thin films show significant redshift and broadening when the Y6 loading ratios exceed 10 wt%, due to various aggregates formation and energy levels splitting. As shown in **Supplementary Fig. 4**, by keeping the FC parameters of the Y6 solution constant (peak 1 and 2), Four more FC progressions at transition energies E_{03} - E_{06} can be found for the 10 wt% and 100 wt% Y6 thin films. Since each progression represents the electronic configuration of isolated molecules and different dimers, the FC parameter should be similar in different samples. During the fitting, the values of the transition energies (E), the vibrational energy ($\hbar\omega$), the Huang-Rhys parameter (S_1), and the Gaussian linewidth parameter (λ) for each

progression were kept almost the same for all samples. Only the peak intensity values (A3 to A6) were adjusted to get the best spectral fit. These fitting results and FC parameters we obtained are very consistent with the results obtained by Kroh et al¹⁶. To clearer to elaborate the fitting procedures, the above discussion has been included in the revised Supporting Information.

(6) When an external field is applied, both the ground state and excited state energies can be altered. Distinguishing the contributions from the ground state and excited states to the measured EA signals is important. The charge transfer characteristics of excited states may indeed play a more significant role in the photon-to-current conversion process. Exploring and discussing the impact of both ground and excited states on the EA signals would provide a comprehensive understanding of the system.

Response:

Thank you for your suggestion. As discussed in **page 4** in the Supplementary Information, according to the Stark effect, the energy of the ground and excited states will be changed under an electrical field (**Eq.3** and **Eq. 4**). However, both the dipole moment and polarizability of the states are electrical field independent. Typically, such independence can be justified by examining the quadratic dependence of the field dependence of the EA signals as shown in the insets below. Regarding to ground state dipole moment, the reported ground state dipole moments for ITIC (0 D) and Y6 (0.7 D) are relatively small,¹⁹ compared to the extracted value from EA (8 D) for the difference in dipole moment between the ground (μ_0) and excited (μ_1) states. Therefore, we can safely ascribe the extracted values to the contribution from the excited state. In case the ground state dipole moment is not negligible, the theoretical calculation should be used to calculate the ground state dipole moment, as the EA signal is dependent on the relative change ($\Delta\mu$) of the dipole moments between the ground and excited states.

Figure R8. The measured second harmonic EA spectra of 10 wt% and 100 wt% Y6 thin films. These data have been used to do the fitting analysis to extract $\Delta\mu$ and Δp at different excitation energies.

(7) In a film sample, multiple configurations of Y6 aggregations can exist, each characterized by different optical responses. It is crucial to consider this effect when interpreting the EA data. Accounting for the presence of different aggregation configurations and their respective optical responses would contribute to a more accurate and comprehensive analysis of the experimental results.

Response:

Thank you and we fully agree with your suggestion. In the revised manuscript, as shown in **Supplementary Fig. 5**, we have included the EA analysis of different energy excitations in dispersed and aggregated Y6 samples. As discussed above, we can identify the contribution of different aggregates and non-aggregates to the optical absorption spectrum by using the Frank-Condon analysis. Region 1 corresponds to S_1 state (1.37 eV – 1.65 eV) and it is mainly contributed by the low energy part of aggregate 2 (Agg. II, LE), region 2 (1.72 eV – 1.93 eV) has more contribution from the non-interacting molecules, whereas region 3 (above 2.03 eV) has contribution from the high energy part of aggregate 2 (Agg. II, HE), which is above the transporting energy level (~ 2 eV)⁷. Accordingly, as shown in **Figure R9** and **Table R4** of the EA fitting results, region 2 has the smallest $\Delta\mu/\Delta p$ value, indicating the weak CT character. Region 1 has a slightly higher $\Delta\mu/\Delta p$ value, indicating an increased CT character. In region 3, the $\Delta\mu/\Delta p$ value is ten times higher than that in region 2, suggesting a very strong CT character. In this region, $\Delta\mu$ is extremely large (26.02 ± 3 Debye) and Δp is significantly reduced. It is possibly because the excitation in region 3 falls into the transporting energy level which has a high probability of free charge generation.

However, it is challenging to perform the EA analysis on individual FC progression obtained in the optical absorption analysis. Mainly because of the large degree of overlap between different progressions. Currently, we use the device absorbance derivatives to perform the EA analysis. The device absorbance spectral characteristic is different from that in thin film due to the optical

interference effect. Therefore, we may need to do detailed ellipsometry measurements on thin films of different loading ratios to obtain the n (refractive index) and k (extinction coefficient) values by fitting the device absorbance data. It would be a very good direction, but It would involve an extensive of work and currently we would like to focus our discussions mainly on the evolution of CT properties of the E_{S1} state from isolated to aggregated Y6 molecules.

Alternatively, we can roughly analyze the contribution from different aggregates by part-by-part fitting of the EA spectra of different samples. The regions selected for part-by-part EA fitting were determined by the contribution of the aggregated (low energy and high energy) and non-aggregated Y6 on the optical absorption spectra.

Figure R9. Electroabsorption spectra fitting. Measured second harmonic (2ω) EA signals (transmission mode) of 10 wt% of Y6 in PVK fitted with Equation 2. The red symbols represent the fitted region of the EA signal, which corresponds to the (a) low energy aggregate (Agg. II, LE), (b) non-interacting molecules, and (c) high energy aggregate (Agg. II, HE), regions of thin

film absorbance of 10 wt% Y6. The extracted $\Delta\mu$ (change in dipole moment) and Δp (change in polarizability) values of different aggregate configurations are shown in the figure.

FCWD		Energy (eV)	$\Delta\mu$ (D)	Δp (10^{-22} cm^3)	$\Delta\mu/\Delta p \times 10^{22} \text{ D/cm}^3$	FE/ CT
Region 1	Agg. II, LE	1.37-1.65	9.31 ± 1.2	2.92 ± 0.5	3.19 ± 0.5	Little CT
Region 2	Non-interacting	1.72-1.93	10.18 ± 0.5	5.77 ± 0.4	1.76 ± 0.1	FE
Region 3	Agg. II, HE	2.03-2.41	26.02 ± 3.2	0.94 ± 0.7	27.68 ± 18.5	CT

Table R4. $\Delta\mu$ and Δp values extracted from EA fitting, their ratio and nature of excitons at different aggregated and non-interacted regions of 10 wt% Y6 in PVK.

Similarly, as shown in **Figure. R10** for the 100 wt% Y6 sample, region 1 (1.31 eV – 1.64 eV) represents the low energy aggregates 1 and 2 (Agg. I & II, LE), region 2 (1.64 eV – 1.87 eV) represents the non-interacting molecules of Y6, whereas region 3 (above 1.97 eV) represents the high energy aggregates (Agg. I & II, HE) as well as the transporting energy level. According to the EA fitting results as shown in summarized in **Table R5**, region 2 has the smallest $\Delta\mu/\Delta p$ value, consistent with the value obtained in the 10 wt% sample. Region 1 has a much larger $\Delta\mu/\Delta p$ value, indicating its strong CT character. It should be noted that region 1 in the 100 wt% Y6 sample consists of Agg. I & II, LE and region 1 in the 10 wt% sample only consists of Agg. II, LE. Therefore, the much higher (3 times) $\Delta\mu/\Delta p$ value in the 100 wt% Y6 sample suggests that the CT character in Agg. I, LE is much stronger than that in Agg. II, LE. Similar to the 10wt% sample, region 3 in the 100 wt% Y6 sample also has a very high $\Delta\mu/\Delta p$ value. Possibly due to its strong CT character from the above transporting energy level excitation.

Figure R10. Electroabsorption spectra fitting. Measured second harmonic (2ω) EA signals (transmission mode) of 100 wt% of Y6 fitted with Equation 2. The red symbols represent the fitted region of the EA signal, which corresponds to the low energy aggregate (Agg. I & II, LE), non-interacting molecules, and high energy aggregate (Agg. I & II, HE), regions of thin film absorbance of pure Y6. The extracted $\Delta\mu$ (change in dipole moment) and Δp (change in polarizability) values of different samples are shown in the figure. The table below shows the extracted $\Delta\mu$ and Δp values from EA fitting, their ratio, and the nature of excitons at different bands of 100 wt% Y6.

FCWD	Energy range (eV)	$\Delta\mu$ (D)	$\Delta\rho$ (10^{-22} cm^3)	$\Delta\mu/\Delta\rho \times 10^{22} \text{ D/cm}^3$	FE/ CT	
Region 1	Agg. I & II, LE	1.31 – 1.64	8.35 ± 0.2	0.92 ± 0.1	9.07 ± 0.3	CT
Region 2	Non-interacting	1.64- 1.87	6.95 ± 0.8	3.49 ± 0.3	1.99 ± 0.2	FE
Region 3	Agg. I & II, HE	1.97- 2.41	13.61 ± 2.2	0.44 ± 0.3	30.93 ± 18.9	CT

Table R5. $\Delta\mu$ and $\Delta\rho$ values extracted from EA fitting, their ratio and nature of excitons at different aggregated and non-interacted regions of 100 wt% Y6.

Although the above analysis still has limitations on clearly separating the contribution from different aggregates due to the considerable spectral overlapping, the results have provided clear differentiation of the contributions on the CT character from aggregated Y6 and non-interacting Y6. More importantly, the EA analysis at different energies further demonstrates the validity of the contribution of $\Delta\rho$ and the proposed $\Delta\mu/\Delta\rho$ ratio on the assessment of the CT character.

References

1. Chin, B. C., Misawa, K., Masuda, T. & Kobayashi, T. Large static dipole moment in substituted polyacetylenes obtained by electroabsorption. *Chem. Phys. Lett.* **318**, 499–504 (2000).
2. Carsten, B. *et al.* Examining the effect of the dipole moment on charge separation in donor-acceptor polymers for organic photovoltaic applications. *J. Am. Chem. Soc.* **133**, 20468–20475 (2011).
3. Bernardo, B. *et al.* Delocalization and dielectric screening of charge transfer states in organic photovoltaic cells. *Nat. Commun.* **5**, 3245 (2014).
4. Chen, X., Chan, C. C., Mahadevan, S., Guo, Y., Zhang, G., Yan, H., Wong, K. S., Yip, H., Bredas, J., Tsang, S. W., & Chow, P. C. Intermolecular CT excitons enable nanosecond excited-state lifetimes in NIR-absorbing non-fullerene acceptors for efficient organic solar cells. *ArXiv:2304.09408* (2023).
5. Zhang, G. *et al.* Delocalization of exciton and electron wavefunction in non-fullerene acceptor molecules enables efficient organic solar cells. *Nat. Commun.* **11**,3943 (2020).
6. Kupgan, G., Chen, X. K. & Brédas, J. L. Molecular packing of non-fullerene acceptors for organic solar cells: Distinctive local morphology in Y6 vs. ITIC derivatives. *Mater. Today Adv.* **11**,100154 (2021).
7. Karuthedath, S. *et al.* Intrinsic efficiency limits in low-bandgap non-fullerene acceptor organic solar cells. *Nat. Mater.* **20**, 378–384 (2021).
8. Siebert-Henze, E. *et al.* Electroabsorption studies of organic p-i-n solar cells: Increase of the built-in voltage by higher doping concentration in the hole transport layer. *Org. Electron.* **15**, 563–568 (2014).
9. Siebert-Henze, E. *et al.* Built-in voltage of organic bulk heterojunction p-i-n solar cells measured by electroabsorption spectroscopy. *AIP Adv.* **4**, 0–10 (2014).
10. Liao, X. *et al.* NIR Photodetectors with Highly Efficient Detectivity Enabled by 2D Fluorinated Dithienopicenocarbazole-Based Ultra-Narrow Bandgap Acceptors. *Adv. Funct. Mater.* **32**, 2204255 (2022).
11. Yao, Z. *et al.* Dithienopicenocarbazole-Based Acceptors for Efficient Organic Solar Cells with Optoelectronic Response Over 1000 nm and an Extremely Low Energy Loss. *J. Am. Chem. Soc.* **140**, 2054-2057 (2018).
12. Wang, R. *et al.* Charge Separation from an Intra-Moiety Intermediate State in the High-Performance PM6:Y6 Organic Photovoltaic Blend. *J. Am. Chem. Soc.* **142**, 12751–12759 (2020).

13. Price, M. B. *et al.* Free charge photogeneration in a single component high photovoltaic efficiency organic semiconductor. *Nat. Commun.* **13**, 2827 (2022).
14. Sağlamkaya, E. *et al.* What is special about Y6; the working mechanism of neat Y6 organic solar cells. *Materials Horizons.* **10**, 1825–1834 (2023).
15. Xiao, Y. *et al.* Unveiling the crystalline packing of Y6 in thin films by thermally induced “backbone-on” orientation. *J. Mater. Chem. A* **9**, 17030–17038 (2021).
16. Kroh, D. *et al.* Identifying the Signatures of Intermolecular Interactions in Blends of PM6 with Y6 and N4 Using Absorption Spectroscopy. *Adv. Funct. Mater.* 2205711 (2022).
17. Gao, J. *et al.* Over 16.7 % efficiency of ternary organic photovoltaics by employing extra PC71BM as morphology regulator. *Sci. China Chem.* **63**, 83–91 (2020).
18. Brédas, J. L., Beljonne, D., Coropceanu, V. & Cornil, J. Charge-transfer and energy-transfer processes in π -conjugated oligomers and polymers: A molecular picture. *Chem. Rev.* **104**, 4971–5003 (2004).
19. Li, T. *et al.* Asymmetric Glycolated Substitution for Enhanced Permittivity and Ecocompatibility of High-Performance Photovoltaic Electron Acceptor. *J. Am. Chem. Soc. Au.* **1**, 1733-1742 (2021).

REVIEWER COMMENTS

Reviewer #3 (Remarks to the Author):

Sorry for the late reply. I had a careful look at all submitted documents. I also read the Nat Energy Submission (now Ref 54). The authors have extended the spectral characterization of the linear optical and electrooptical characterization by performing a detailed decomposition of the absorption spectra for different Y6 loading in the framework of the MLJ-theory, which reveals a continuous transition from completely dissolved Y6 molecules to fully aggregated Y6 clusters with two types of aggregates. This information is then used for a more detailed analysis of the EO signal in different regions in terms of the $\Delta\mu/\Delta p$ value of the different species (aggregate I and II, amorphous phase, transport levels). With this, the authors provide a more detailed inside into them nature of the different species which compose the absorption of Y6 in different states of aggregation.

While I have no doubt that this inside is useful for the community to understand optical and electrooptical properties of Y6 in its differently aggregated states, I still miss a quantitative correlation which the molecular properties. The authors state in page 13: "In our previously reported DFT calculation results, Y6 dimers show reduced wavefunction overlapping parameter and oscillator strength for all three different dimer configurations as compared to that in monomer⁵⁴. Supplementary Table 3 shows a comparison between the overlapping parameter between the electron-hole wavefunctions (O_{e-h}) calculated from DFT and the change in polarizability (Δp) extracted from EA analysis. In fact, an increase in polarizability of a state with more delocalized electron and hole wavefunctions might increase the chance of wavefunction overlapping and recombination. Thus, we propose that the CT characteristics can be more generally justified b considering the $\Delta\mu/\Delta p$ ratio". These arguments are certainly reasonable but why didn't the authors approach Prof. Bredas for the values of $\Delta\mu$ and Δp for the monomer and the different dimers to arrive at a molecular interpretation of these two properties in relation to the type of aggregation. These values would be readily available from their DFT calculations. I am also sure that the group of Prof. Bredas would be able to provide the directional properties of $\Delta\mu$ and Δp (in relation to the molecular alignment, which would help to understand the DC bias dependent EA data.

As it is now, the paper is not suited for publication in Nat. Comm.

Reviewer #4 (Remarks to the Author):

In response to my previous question, the authors explain that since only CF solvents are used in this paper, the effects of solvents have not been studied, but this problem has been noted and should be

studied in depth. In fact, solvents have a significant impact on molecular buildup. In addition, authors should consider the issue of universality for emerging materials.

The authors explain why choosing the MD simulation results is more relevant to our films which were prepared by spin-coating. Does spin-coating affect the molecular stacking of Y6 or ITIC? What about the thickness? The author needs further explanation of this issue.

The authors said that other insulating polymers PS and PMMA are used, so what is the reason for choosing these materials? The author should give a further explanation.

Besides, in the "Optical absorption of Y6 and ITIC in solution and thin films" section, the authors use nm as the unit in the description of the peak position while eV in Figure 1, the author should unify the format.

Reviewer #5 (Remarks to the Author):

The authors have well addressed my concerns.

Response to the reviewers

Many thanks to the insightful and constructive comments of the reviewers. The changes have been highlighted in the revised manuscript, and the detailed replies to the comments are as follows.

Reviewer #3:

In fact, an increase in the polarizability of a state with more delocalized electron and hole wavefunctions might increase the chance of wavefunction overlapping and recombination. Thus, we propose that the CT characteristics can be more generally justified by considering the $\Delta\mu/\Delta p$ ratio". These arguments are certainly reasonable but why didn't the authors approach Prof. Bredas for the values of $\Delta\mu$ and Δp for the monomer and the different dimers to arrive at a molecular interpretation of these two properties in relation to the type of aggregation? These values would be readily available from their DFT calculations. I am also sure that the group of Prof. Bredas would be able to provide the directional properties of $\Delta\mu$ and Δp (in relation to the molecular alignment, which would help to understand the DC bias-dependent EA data.

Response:

Thanks for your kind effort in reviewing our manuscript again. We appreciate your constructive comments. Following the reviewers' suggestion, we established a collaboration with the Prof. Brédas group, and they showed great interest in our EA experimental results. They also agreed that this study would be more systematic and quantitative if the extracted $\Delta\mu$ and Δp values from the EA fitting and proposed model could be verified using DFT calculations. We are excited that the calculation results are in good agreement with our experimental results. The theoretical study also provides an in-depth physical interpretation of the change in excitonic parameters and the experimentally derived correlation between $\Delta\mu/\Delta p$ ratio and CT characters in Y6. It is found that the strong CT character in Y6 is promoted by the stabilization of the CT energy upon aggregation. And the similar $\Delta\mu$ values between the isolated and aggregated Y6 molecules observed in the EA measurement are attributed to the cancellation effect of the dipole moment

from intra-molecular and inter-molecular CT transitions ($\Delta\mu_{intra}$ and $\Delta\mu_{inter}$). A detailed description of the calculation method and results are highlighted in the revised manuscript which is briefly described as below:

They performed excited-state calculations using the structure of molecular dimers extracted from crystal structures to explain the EA results and to quantify the CT character of Y6^{1,2}. In the calculations, the long-range corrected ω B97X-D functional and 6-31G (d, p) basis set was used, where the range-separation parameter (ω) was set at 0.0126 Bohr⁻¹, as previously report³. All the DFT calculations were performed with the Gaussian 16 package⁴.

As discussed earlier, we fitted the EA_{2 ω} spectra of devices with different loading ratios of Y6 at its first optical transition (S_1 state). In previous reports, the strong CT character observed in some organic systems has usually been associated with the large $\Delta\mu$ values determined by EA⁵⁻⁷. Due to strong second derivative features in the EA_{2 ω} spectra of pure Y6, we expect a larger $\Delta\mu$ value. Surprisingly, the $\Delta\mu$ value in pure Y6 film (8.35 D) is similar, or even smaller than, that in the 1 wt% and 10 wt% Y6 film (~9.5 D). In contrast, Δp value is reduced by a factor of 3 upon increasing the loading ratio, from 290 Å³ in the 1 wt% Y6 device to 95 Å³ in the pure Y6 device. Thus, we found that the large increase in CT in clustered Y6 molecules is not due to a large increase in excited-state dipole moment, $\Delta\mu$, as observed in other organic systems, but due to the reduced polarizability change, Δp . Instead of using only $\Delta\mu$ or Δp value, we must use $\Delta\mu/\Delta p$ to quantify the CT characteristics of Y6. $\Delta\mu/\Delta p$ ratio increases by a factor of 3 from 0.03 D/Å³ in 1 wt% and 10 wt% Y6 to 0.09 D/Å³ in pure Y6. On the other hand, the $\Delta\mu/\Delta p$ ratio in the case of ITIC remains the same, whether a diluted ITIC: PVK film or a neat ITIC film is considered, indicating a strong LE (Frenkel) character.

To explain the EA_{2 ω} data and to quantify the CT character of the Y6 excitations, we performed excited-state calculations using molecular dimers extracted from the crystal structures^{1,2}. Interestingly, it is found that Y6 contains two monomers, **M1** and **M2** (helical and bent conformations), which were overlooked in previous discussions on Y6 and form different dimer configurations as reported in previous literature. The energy of the S_1 states in bent and twisted monomers differ by about 300 meV (see **Figure 1**). The DFT calculations show that the

average $\Delta\mu$ value of the Y6 molecule value related to $S_0 \rightarrow S_1$ excitation is about 8 D, consistent with the EA data. The relatively large value according to natural transition orbitals (NTOs) can be attributed to the intra-molecular CT character of $S_0 \rightarrow S_1$ excitation in Y6 monomers as shown in **Figure 1**. The average Δp value of the Y6 monomer is estimated to be about 800 \AA^3 which is to some extent larger than the experimental value (300 \AA^3).

Figure 1. The natural transition orbitals (NTOs) of the twisted (M_1) and bent (M_2) monomers and the dimer formed from these two monomers. Here, E denotes the transition energy in eV, f represents the oscillator strength, λ is the eigenvalue of the NTO, and $\Delta\mu$ signifies the dipole moment change in Debye for the $S_0 \rightarrow S_1$ transition.

The presence of two different groups of monomers resulted in two groups of dimers, *i.e.* symmetric and asymmetric. The calculations show that $S_0 \rightarrow S_1$ excitation in all dimers has a significant inter-molecular CT character. However, the CT contribution is in particular large in asymmetric dimers. **Figure 1** displays the NTOs of an asymmetric dimer where the CT and LE contributions are nearly equal. However, asymmetric dimers with nearly 100% CT contribution exist as well (dimer C), as shown in **Figure 2**.

- DFT results for different dimer configurations of Y6.

Figure 2. Illustration of two distinct dimer configurations of Y6, labelled as A and B, extracted from the crystal structure reported from our previous work¹ and Dimer C (which closely resembles Dimer A) retrieved from the work by Marks and co-workers². Both dimers A and C comprise asymmetric constituent monomers, whereas dimer B features symmetric constituent monomers. We replaced the long side chains with $-CH_3$ groups for our calculations.

Detailed information of different dimer configurations (Dimer A, B, and C) of Y6 and calculated physical properties such as oscillator strength (f), change in polarizability ($\Delta\rho$), and change in dipole moment ($\Delta\mu$) at different excitation energies (ΔE) are given below in **Figure 3 (a,b,c)** and **Table 1 (a,b,c)**.

Figure 3.a. Natural Transition Orbitals (NTOs) representing the hole and electron distributions in the lowest four singlet excited states of dimer A of Y6.

Table 1.a. Calculated energies for the lowest four singlet excited states (S_1 to S_4 , denoted as ΔE) of dimer A of Y6, along with the corresponding oscillator strengths (f) for $S_0 \rightarrow S_n$ ($n=1$ to 4) transitions. The table also includes the variations in dipole moment ($\Delta\mu$) and polarization ($\Delta\rho$) during these transitions. Additionally, similar values for the S_1 state are provided for the constituent monomeric units (M1 and M2), comprising the dimer A of Y6.

	S_1	S_2	S_3	S_4	M_1	M_2
ΔE (eV)	1.65	1.83	1.86	2.03	2.06	1.75
f	0.86	1.27	0.05	0.27	2.15	2.03
$\Delta\mu$ (D)	9.80	13.44	13.58	17.24	4.72	8.38
$\Delta\rho$ (\AA^3)	1296	599	749	-973	864	690

Figure 3.b. Natural Transition Orbitals (NTOs) represent the hole and electron distributions in the lowest four singlet excited states of dimer B of Y6.

Table 1.b. Calculated energies for the lowest four singlet excited states (S_1 to S_4 , denoted as ΔE) of dimer B of Y6, along with the corresponding oscillator strengths (f) for $S_0 \rightarrow S_n$ ($n=1$ to 4) transitions. The table also includes the variations in dipole moment ($\Delta\mu$) and polarization ($\Delta\rho$) during these transitions. Additionally, similar values for the S_1 state are provided for the constituent monomeric units (M1 and M2), comprising the dimer B of Y6.

	S_1	S_2	S_3	S_4	M_1	M_2
ΔE (eV)	1.63	1.63	1.69	1.75	1.75	1.75
f	0.02	0.39	0.00	3.62	2.03	2.03

$\Delta\mu$ (D)	10.59	10.37	0.15	0.11	8.41	8.41
Δp (\AA^3)	59449	-58433	891	-552	687	687

Figure 3.c. Natural Transition Orbitals (NTOs) representing the hole and electron distributions in the lowest four singlet excited states of dimer C of Y6.

Table 1.c. Calculated energies for the lowest four singlet excited states (S_1 to S_4 , denoted as ΔE) of dimer C of Y6, along with the corresponding oscillator strengths (f) for $S_0 \rightarrow S_n$ ($n=1$ to 4) transitions. The table also includes the variations in dipole moment ($\Delta\mu$) and polarization (Δp) during these transitions. Additionally, similar values for the S_1 state are provided for the constituent monomeric units (M_1 and M_2), comprising the dimer C of Y6.

	S_1	S_2	S_3	S_4	M_1	M_2
ΔE (eV)	1.60	1.70	1.78	1.90	1.75	1.81
f	0.137	1.033	1.922	0.201	2.10	1.82
$\Delta\mu$ (D)	30.41	17.05	10.86	26.08	12.21	7.42
Δp (\AA^3)	722	203	1180	576	738	1255

The $\Delta\mu$ and Δp values for monomer and dimers extracted from EA fitting have some correlation with that from the theoretical calculations. However, expecting exactly similar values from the two approaches is somewhat impractical. For EA experiment, we extracted the excitonic parameters ($\Delta\mu$, Δp) of Y6 at different loading ratios in thin films and the values can be an ensemble average of all different dimer configurations. In DFT, we calculated the $\Delta\mu$ and Δp

values from the interaction between two adjacent molecules. For Y6 system, $\Delta\mu$ originates from both intra-molecular and inter-molecular CT transitions ($\Delta\mu_{intra}$ and $\Delta\mu_{inter}$). In amorphous Y6 films, there will be a distribution of $\Delta\mu_{intra}$ and $\Delta\mu_{inter}$ parameters, in some cases $\Delta\mu_{intra}$ and $\Delta\mu_{inter}$ can be cancelled out or act in a concerted manner. Furthermore, the presence of two types of monomers, their molecular orientation, vibronic coupling, and wavefunction delocalization are some of the factors that can complicate the analysis of calculating values of $\Delta\mu$ and Δp .

To shed more light on this issue we used a four-state model involving the two LE and two CT excitations of a dimer as discussed above. This model was successfully used before to rationalize the EA properties in organic molecular dimers^{8,9}.

$$H = \begin{pmatrix} E_{M_1}^{LE} & d & t_1 & t_2 \\ d & E_{M_2}^{LE} & t_2 & t_1 \\ t_1 & t_2 & E_{CT}^{M_1^+M_2^-} & 0 \\ t_2 & t_1 & 0 & E_{CT}^{M_1^-M_2^+} \end{pmatrix}$$

In this matrix, $E_{M_1}^{LE}$ and $E_{M_2}^{LE}$ are the LE (Frenkel) excitation energies for monomer 1 (M_1) and monomer 2 (M_2) within the dimer, respectively; $E_{CT}^{M_1^+M_2^-}$ and $E_{CT}^{M_1^-M_2^+}$ represent the charge transfer energies for excitation from M_1 to M_2 or vice versa. The effect of the electric field (F) is described as follows:

$$E_{M_1}^{LE} = E_{M_1}^{LE}(0) - \Delta\mu_{M_1}^{LE} \times F - 0.5 \times \Delta p_{M_1}^{LE} \times F^2 \quad (24)$$

$$E_{M_2}^{LE} = E_{M_2}^{LE}(0) - \Delta\mu_{M_2}^{LE} \times F - 0.5 \times \Delta p_{M_2}^{LE} \times F^2 \quad (25)$$

$$E_{CT}^{M_1^+M_2^-} = E_{CT}^{M_1^+M_2^-}(0) - \Delta\mu_{CT}^{M_1^+M_2^-} \times F - 0.5 \times \Delta p_{CT}^{M_1^+M_2^-} \times F^2 \quad (26)$$

$$E_{CT}^{M_1^-M_2^+} = E_{CT}^{M_1^-M_2^+}(0) - \Delta\mu_{CT}^{M_1^-M_2^+} \times F - 0.5 \times \Delta p_{CT}^{M_1^-M_2^+} \times F^2 \quad (27)$$

Based on previous DFT calculations^{1,10} we used the following model parameters: $\Delta\mu_{M_1}^{LE} = \Delta\mu_{M_2}^{LE} = 8$ D; $\Delta\mu_{CT}^{M_1^+M_2^-} = \Delta\mu_{CT}^{M_1^-M_2^+} = 10$ D; $\Delta\mu_{CT}^{M_1^-M_2^+} = \Delta p_{CT}^{M_1^-M_2^+} = 0$; $\Delta p_{M_1}^{LE} = \Delta p_{M_2}^{LE} = 950$ Å³; $E_{CT}^{M_1^+M_2^-}(0) - E_{CT}^{M_1^-M_2^+}(0) = 50$ meV; $d = 50$ meV; and $t_1 = t_2 = 100$ meV. The energies of the

eigenstates of the four-state model were calculated for ΔE_{CT-LE} values in the range from 0.3 eV to -0.3 eV as a function of the electric field. The $\Delta\mu$ and Δp values were then derived by computing the first and second derivatives of the S_1 energy as a function of the electric field.

Using the results generated by the four-state model, we computed the dependence of $\Delta\mu$ and Δp values of a Y6 dimer as a function of energy separation between LE and CT states (ΔE_{LE-CT}), these results are shown in **Figure 4**. We note that here, for the sake of simplicity, we fixed the values of other values for the microscopic parameters that, strictly speaking, are also distance-dependent. As seen from **Figure 4a**, as the CT state energy is stabilizing (ΔE_{CT-LE} decreases) the $\Delta\mu$ increases while Δp decreases, as a consequence $\Delta\mu/\Delta p$ increases as well. The calculations also show (**Figure 4b**) that as ΔE_{CT-LE} decreases the CT contribution to the S_1 (CT%) continuously increases as expected. As the energy of CT excitation is located below that of the LE excitation ($\Delta E_{CT-LE} < 0$), a sharp increase with the increase of CT% parameter is also observed for the $\Delta\mu/\Delta p$. This supports our findings that $\Delta\mu/\Delta p$ can be used as a figure of merit to explain the strong CT character of Y6.

Figure 4. Correlation of $\Delta\mu$ and Δp with charge transfer weight (CT%) of the dimer S_1 excitation, as calculated from the four-state model. (a) Variations in $\Delta\mu$ and Δp values of Y6 dimers (black and red solid lines, respectively) as a function of the energy difference between the LE and CT states (ΔE_{CT-LE}). (b) Comparison of the ratio of $\Delta\mu/\Delta p$ (red solid line) to the charge transfer weight (black solid line) vs. the energy separation between the LE and CT states (ΔE_{CT-LE}).

- **DFT results of different dimer configurations of ITIC.**

To further support our findings, we also verified the excitonic properties of ITIC. From EA analysis, we found that $\Delta\mu$ and Δp values are almost similar, indicating the presence of strong Frenkel excitons. Interestingly, the DFT calculations indicate that the dimers in ITIC crystals have a pure LE character (see **Figures 5 and 6** and **Table 2** below).

Figure 5. Illustration of two distinct dimer configurations of ITIC, labelled as A and B, extracted from the crystal structure reported in the literature¹¹. Both dimers have symmetric constituent monomers. We replaced the long side chains with $-\text{CH}_3$ groups for our calculations.

Figure S6.a. Natural Transition Orbitals (NTOs) representing the hole and electron distributions in the lowest singlet excited state of dimer A of ITIC.

Table 2.a. Calculated energies for the lowest four singlet excited states (S_1 to S_4 , denoted as ΔE) of dimer A of ITIC, along with the corresponding oscillator strengths (f) for $S_0 \rightarrow S_n$ ($n=1$ to 4) transitions. The table also includes the variations in dipole moment ($\Delta\mu$) and polarization (Δp) during these transitions. Similar values for the S_1 state are also provided for the constituent monomeric units (M_1 and M_2), comprising the dimer A of ITIC.

	S_1	S_2	S_3	S_4	M_1	M_2
ΔE (eV)	1.85	1.86	2.10	2.10	1.86	1.86
f	5.66	0.01	0.00	0.00	2.80	2.81
$\Delta\mu$ (D)	0.27	0.05	100.91	100.55	0.07	0.17
Δp (\AA^3)	1236	925	1765	1747	1063	10.59

Figure S6.b. Natural Transition Orbitals (NTOs) represent the hole and electron distributions in the lowest singlet excited states of dimer B of ITIC.

Table S2.b. Calculated energies for the lowest four singlet excited states (S_1 to S_4 , denoted as ΔE) of dimer B of ITIC, along with the corresponding oscillator strengths (f) for $S_0 \rightarrow S_n$ ($n=1$ to 4) transitions. The table also includes the variations in dipole moment ($\Delta\mu$) and polarization (Δp) during these transitions.

Additionally, similar values for the S_1 state are provided for the constituent monomeric units (M1 and M2), comprising the dimer B of ITIC.

	S_1	S_2	S_3	S_4	M_1	M_2
ΔE (eV)	1.78	1.84	1.95	1.99	1.86	1.83
f	3.86	1.59	0.17	0.16	2.81	2.97
$\Delta\mu$ (D)	5.36	8.44	48.80	44.73	0.07	0.16
Δp (\AA^3)	1745	2737	1414	-926	1059	869

In summary, we used excited state calculations to correlate the $\Delta\mu$ and Δp values for monomer and dimers extracted from EA fitting analysis. Some of the key findings from the theoretical calculations are:

- Y6 has two kinds of monomers based on their conformation (helical and bent), which have been overlooked in previous reports.
- The average change in dipole moment value at its first optical transition is about 8 D, which agrees with the EA data.
- These two sets of monomers lead to the formation of symmetric and asymmetric dimers in which all dimers around the S_1 state show a significant intermolecular CT character.
- Dimers with pure CT excitations have a lesser change in polarizability.
- Using the four-state model, we found a correlation between the $\Delta\mu/\Delta p$ ratio and strong CT characteristics of Y6, which was proposed earlier from our EA experimental results.
- DFT calculations of ITIC underline that the dimers have a strong Frenkel excitonic nature.

It should be noted that finding a one-to-one correlation between DFT data and experimental results may be challenging. To get deeper insights, we suggest using a proper statistical analysis to calculate a few hundred dimers using MD-generated morphology, however, which would take at least 3 to 4 months for a single run. We consider it an opportunity to continue our investigation on Y6 and share more insightful information in our future work.

Reviewer #4 (Remarks to the Author):

1) In response to my previous question, the authors explain that since only CF solvents are used in this paper, the effects of solvents have not been studied, but this problem has been noted and should be studied in depth. In fact, solvents have a significant impact on molecular buildup. In addition, authors should consider the issue of universality for emerging materials.

Response:

Thanks for your constructive comment and kind effort in reviewing our manuscript. As you mentioned, solvents have a prominent effect on molecular aggregation. James Durrant and co-workers have recently reported that Y6 in CB solvent has edge-on orientation in thin films, whereas CF gives face-on orientation¹². When considering the solubility and miscibility of Y6 and PVK, CF is the best candidate among other solvents. CF is the solvent used in most high-performing OPV devices based on PM6 and Y6. Annealing is also required in thin films using CB solvent. As shown in **Figure 1**, Y6 and ITIC thin films prepared in CB solvent exhibit a more red-shifted absorption spectrum when compared to that in CF solvent. Y6 in CB shows spectral narrowing, possibly due to its molecular packing and orientation difference. CB solvent induces strong molecular aggregation due to its low volatility and annealing process. We are still investigating the effect of solvents, solvent additives, annealing, and polymer matrix on the intermolecular interactions of Y6 molecules. Hopefully, we can share them in our next work.

Figure 1. The normalized absorption spectrum of thin films with a) Y6 and b) ITIC dissolved in different solvents such as CF and CB. Quartz was used as the substrate.

2) The authors explain why choosing the MD simulation results is more relevant to our films which were prepared by spin-coating. Does spin-coating affect the molecular stacking of Y6 or ITIC? What about the thickness? The author needs further explanation of this issue.

Response:

Thanks for your comment. We tried to fabricate Y6 thin films of varied thickness from 100 to 150 nm. As per the EA fitting equation (1), difference thicknesses do not affect the excitonic parameters.

$$\frac{\Delta T}{T} \frac{1}{2\omega} = \left[\frac{1}{4} \frac{\partial A_D}{\partial E} (\Delta p) + \frac{1}{12} \frac{\partial^2 A_D}{\partial E^2} (\Delta \mu^2) \right] \frac{V_{ac}^2}{0.43d^2} \sin \left[2\omega t + \frac{\pi}{2} \right] \quad (1)$$

Since the electric field for the EA measurement was confined to 10^4 to 10^5 V/cm, we optimized the film thickness of pure Y6 around 100 to 150 nm. As shown in **Figure 2**, Y6 spin-coated on the quartz substrate (CB solvent) with 1000 rpm shows higher absorbance when compared to that with 2000 rpm. We didn't observe any spectral changes in the absorption spectrum of thin films

with varied thicknesses. Figure 3 shows the EA spectrum of 100wt% Y6 thin films with varied thicknesses, say, 80 nm, 131 nm, and 150 nm. The extracted $\Delta\mu$ and $\Delta\rho$ values are consistent. Therefore, our spin-coating approach in this study does not affect the molecular packing. There are no research reports on thermally evaporated Y6 thin films. Hopefully, we can extend our future research study in this direction.

Figure 2. The optical absorption spectrum of Y6 spin-coated on thin films (CB solvent) with different spinning parameters (1000 and 2000 rpm @30s).

Figure 3. Second harmonic EA spectrum of 100wt% Y6 thin films with different thicknesses (80 nm, 131 nm, and 150 nm) with the same applied AC bias of 5 V.

3) The authors said that other insulating polymers, PS and PMMA, are used, so what is the reason for choosing these materials? The author should give a further explanation.

Response:

Thanks for your comment. To study the intra- and intramolecular charge transfer properties of Y6, we needed to disperse or isolate Y6 molecules in a host polymer. PS, PVK, and PMMA are commonly used host polymers to fabricate solid-solvation thin films with type-I heterojunctions. Those materials are easily available and cost-effective as well. Among these polymers, only PVK can well-disperse Y6 molecules (shown in **Figure S2**). Figure 4 shows the HOMO and LUMO energy levels of Y6 and PVK.

Figure 4. Energy levels of PVK and Y6.

4) Besides, in the “Optical absorption of Y6 and ITIC in solution and thin films” section, the authors use nm as the unit in the description of the peak position, while eV in Figure 1, the author should unify the format.

Response:

Thanks for the suggestion. We have corrected it in the revised manuscript.

Reviewer #5 (Remarks to the Author):

The authors have well addressed my concerns.

Response:

Thanks a lot for your time and effort in reviewing our manuscript again. Your insightful comments on Franck-Condon analysis helped us to improve our work's quality.

References:

1. Zhang, G. *et al.* Delocalization of exciton and electron wavefunction in non-fullerene acceptor molecules enables efficient organic solar cells. *Nat. Commun.* **11**, 3943 (2020).
2. Zhu, W. *et al.* Crystallography, Morphology, Electronic Structure, and Transport in Non-Fullerene/Non-Indacenodithienothiophene Polymer: Y6 Solar Cells. *J. Am. Chem. Soc.* **142**, 14532–14547 (2020).
3. Kupgan, G., Chen, X.K. & Brédas, J.L. Molecular packing of non-fullerene acceptors for organic solar cells: Distinctive local morphology in Y6 vs. ITIC derivatives. *Mater. Today Adv.* **11**, 100154 (2021).
4. Frisch, M. J., G. W. Trucks, H. B. Schlegel, G. E. Scuseria, M, A. Robb, J. R. Cheeseman, G. Scalmani, et al. Gaussian 16 Rev. C. 01, Wallingford, CT (2016).
5. Chin, B. C., Misawa, K., Masuda, T. & Kobayashi, T. Large static dipole moment in substituted polyacetylenes obtained by electroabsorption. *Chem. Phys. Lett.* **318**, 499–504 (2000).
6. Carsten, B. *et al.* Examining the effect of the dipole moment on charge separation in donor-acceptor polymers for organic photovoltaic applications. *J. Am. Chem. Soc.* **133**, 20468–20475 (2011).
7. Bernardo, B. *et al.* Delocalization and dielectric screening of charge transfer states in organic photovoltaic cells. *Nat. Commun.* **5**, 3245 (2014).
8. Petelenz, P., Mixing of Frenkel excitons with charge transfer states in the neighbourhood of a charged defect, *Chemical Physics Letters*, **47** (3), 603-605 (1977).
9. Petelenz, P., Theoretical models for electro-absorption spectroscopy, *Organic Electronics*, **5**, (1–3), 115-127 (2004).
10. Price, M. B. et al. Free charge photogeneration in a single component high photovoltaic efficiency organic semiconductor. *Nat. Commun.* **13**, 2827 (2022).
11. Aldrich, T.J., Matta, M., Zhu, W., Swick, S.M., Stern, C.L., Schatz, G.C., Facchetti, A., Melkonyan, F.S., Marks, T.J., Fluorination Effects on Indacenodithienothiophene Acceptor Packing and Electronic Structure, End-Group Redistribution, and Solar Cell Photovoltaic Response. *J. Am. Chem. Soc.* **141** (7), 3274-3287 (2019).
12. Fu, Y., Lee, T.H., Chin, YC. *et al.* Molecular orientation-dependent energetic shifts in solution-processed non-fullerene acceptors and their impact on organic photovoltaic performance. *Nat Commun* **14**, 1870 (2023).

REVIEWERS' COMMENTS

Reviewer #3 (Remarks to the Author):

I was pleased that the authors followed my recommendation to include results of DFT calculations by the group of JL Bredas. On the other hand, the rather qualitative section on the dipole orientation in thin films has been removed.

With these changes, the paper is of quality and represents a coherent picture of the nature of singlet excitations in non-aggregated and aggregated Y6. Congratulations. I recommend the publication of this manuscript as it is

Reviewer #4 (Remarks to the Author):

The authors have well addressed all my questions.